# EXPRESSIVITY OF NEURAL NETWORKS WITH RANDOM WEIGHTS AND LEARNED BIASES

**Ezekiel Williams**
Mathematics and Statistics
Université de Montréal
ezekiel.williams@mila.quebec

**Alexandre Payeur**
Mathematics and Statistics
Université de Montréal
alexandre.payeur@mila.quebec

**Avery Hee-Woon Ryoo**
Computer Science
Université de Montréal
hee-woon.ryoo@mila.quebec

**Thomas Jiralerspong**
Computer Science
Université de Montréal
thomas.jiralerspong@mila.quebec

**Matthew G Perich**
Neuroscience
Université de Montréal
matthew.perich@umontreal.ca

**Luca Mazzucato**[*]
Biology, Physics, and Mathematics
University of Oregon
lmazzuca@uoregon.edu

**Guillaume Lajoie**[*]
Mathematics and Statistics
Université de Montréal
guillaume.lajoie@mila.quebec

## ABSTRACT

Landmark universal function approximation results for neural networks with trained weights and biases provided the impetus for the ubiquitous use of neural networks as learning models in neuroscience and Artificial Intelligence (AI). Recent work has extended these results to networks in which a smaller subset of weights (e.g., output weights) are tuned, leaving other parameters random. However, it remains an open question whether universal approximation holds when only *biases* are learned, despite evidence from neuroscience and AI that biases significantly shape neural responses. The current paper answers this question. We provide theoretical and numerical evidence demonstrating that feedforward neural networks with fixed random weights can approximate any continuous function on compact sets. We further show an analogous result for the approximation of dynamical systems with recurrent neural networks. Our findings are relevant to neuroscience, where they demonstrate the potential for behaviourally relevant changes in dynamics without modifying synaptic weights, as well as for AI, where they shed light on recent fine-tuning methods for large language models, like bias and prefix-based approaches.

## 1 INTRODUCTION

The universal approximation theorems Hornik et al. (1989); Funahashi (1989); Hornik (1991) of the late 1900s highlighted the *expressivity* of neural network models, i.e. their ability to approximate or *express* a broad class of functions through the tuning of weights and biases, heralding the central role that neural networks play in Machine Learning (ML) and neuroscience today. Since these foundational studies, a rich literature has explored the limits of this expressivity by finding smaller parameter subsets that, when optimized, can still support the approximation of wide classes of functions or dynamics. Prior work has explored the approximation capabilities of Feedforward

---

[*]co-senior authors

Neural Networks (FNNs) and Recurrent Neural Networks (RNNs) where only the output weights are trained Rosenblatt et al. (1962); Rahimi & Recht (2008); Ding et al. (2014); Neufeld & Schmocker (2023); Jaeger & Haas (2004); Sussillo & Abbott (2009); Gonon et al. (2023); Hart et al. (2021), and deep FNNs where only subsets of parameters Rosenfeld & Tsotsos (2019), normalization parameters Burkholz (2023); Giannou et al. (2023), or binary masks–either over units or parameters–are trained Malach et al. (2020).

In this work, we study the expressivity of neural networks where only biases–which can be interpreted as constant inputs to neural units–are learned. While this may seem like an odd pursuit, bias-related learning is central to several active areas of research. In AI, modern sequence models such as transformers can have their outputs reshaped based on examples or instructions presented in an unchanging prefix sequence. No changes of weights are performed in this case, only distinct inputs carrying information about the context are passed to pre-trained but fixed neural network components Li & Liang (2021); Marvin et al. (2023); Garg et al. (2022); Von Oswald et al. (2023). Even more closely related, training only biases is a strategy that has been used recently Zaken et al. (2021) for more efficient fine-tuning. In neuroscience, there is growing evidence that animals can leverage inputs–via long-range projections from higher cortical areas or neuromodulatory nuclei– in order to rapidly and flexibly adapt network dynamics to multiple tasks Mazzucato et al. (2019); Ogawa et al. (2023); Perich et al. (2018); Remington et al. (2018). These tonic inputs mediating the effects of projections from other brain areas to the given, local, circuit are typically modelled as biases (see Supplementary Table 1). Consequently, input-based learning is a conceptually crucial yet poorly understood component of both modern AI systems as well as of brain functions.

Quantifying the expressivity of bias learning would show the degree to which the brain or neural networks can rely on the adaptation of bias-related parameters to structure their dynamics for new tasks, thus providing critical theoretical grounding to this growing literature. If tuning the biases of a neural network will only span a reduced set of functions, or output dynamics, then this would solidify the role of synaptic plasticity as *the* critical component in biological and artificial learning. Conversely, if one can express arbitrary dynamics solely by changing biases, this would motivate deeper investigation of when and how non-synaptic mechanisms might shoulder some of the effort of learning. In this paper, we take a first step towards characterizing the expressivity of bias learning by studying the arguably worst-case scenario of a neural network with unstructured weights. In a regime where all weight parameters are randomly initialized and frozen, and only hidden-layer biases are optimized–which we term *bias learning*–we give theoretical guarantees demonstrating that:

1. bias-learning FNNs with wide hidden layers are universal function approximators with high probability;
2. bias-learning RNNs with wide hidden layers can arbitrarily approximate finite-time trajectories from smooth dynamical systems with high probability.

We further provide empirical support for, and a deeper interrogation of, these results with numerical experiments exploring multi-task learning, motor-control, and dynamical system forecasting.

## 1.1 RELATED WORKS

**Machine Learning**. Many efforts have explored neural networks that are trained to quickly meta-learn new tasks via dynamics in activation space alone, without any adaptation of weights (see Feldkamp et al. (1997); Klos et al. (2020); Cotter & Conwell (1990; 1991); Hochreiter et al. (2001); Subramoney et al. (2024)). Like our work, this research proposes a mechanism by which a network might "learn" any new task without changing weights. However, prior work differs from the current study in that it requires an initial meta-training of all parameters in a network, weights included, before operating in the "fast learning" regime where network variables maintain context information that allow the networks to rapidly adapt to new tasks. In some cases, these context variables can be thought of as biases Cotter & Conwell (1991).

From a mathematical perspective, our work is closely related to masking, particularly the *Strong Lottery Ticket Hypothesis* (SLTH). This hypothesis conjectures that a desired network parameterization could be found as a sub-network in a larger, appropriately-initialized, network Ramanujan et al. (2020). SLTH is typically formulated with respect to weights, i.e., a subnetwork is defined

by deleting weights from the full network. However, a few studies have investigated SLTH where subnetworks are constructed by deleting units Malach et al. (2020), which we term *SLTH over units*. While our study is different for its focus on function approximation via bias optimization, rather than finding "lottery ticket" subnetworks, a key step in our analytic derivations relies on masking in a fashion analogous to proofs of SLTH over units. Thus, our work also provides two results that may be of interest to the SLTH theory: a novel proof of SLTH over units in single-layer FNNs, complementing the work of Malach et al. (2020) (see Connections with Malach et al. 2020 in §B.2 for details), and a first proof of SLTH for RNNs (see Section §2 for more details). Flavours of the lottery ticket hypothesis for RNNs have been explored empirically Yu et al. (2019); García-Arias et al. (2021); Schlake et al. (2022) but we have not encountered its proof, neither over weights nor units, in the literature.

**Neuroscience**. Changes in input biases, which mediate a change in the input-output transfer functions of neurons, can explain the context-dependent effects of expectation Mazzucato et al. (2019), movements Wyrick & Mazzucato (2021) and arousal Papadopoulos et al. (2024) on sensory processing across modalities and brain areas. Some of these effects occur by plasticity-driven changes in amygdalar projections to cortex, induced by associative learning Vincis & Fontanini (2016); Haley et al. (2020). Changes in neural firing threshold and network inputs, similar to bias modulations, were shown to shape network dynamics: threshold heterogeneity can improve network capacity Gast et al. (2024; 2023) and reconfigure circuit dynamics on fast timescales Perich et al. (2018); Remington et al. (2018). A recent study showed that, in RNNs trained to perform neuroscience tasks, learning biases via language model embedding leads to zero-shot generalization to new tasks Riveland & Pouget (2024). Within the reservoir computing approach to modelling in neuroscience, where recurrent weights are random and fixed, bias modulations can toggle between multiple phases (including fixed point, chaos, and multistable regimes) and, strikingly, enable RNN multi-tasking in the absence of any parameter optimization Ogawa et al. (2023). While slightly different than bias parameters, a repertoire of dynamical motifs can also be generated in RNN reservoirs with dynamic feedback loops Logiaco et al. (2021) and by modulating inputs in pre-trained networks Driscoll et al. (2024).

## 2 THEORY RESULTS

### 2.1 FEEDFORWARD NEURAL NETWORKS

This section studies the single-layer FNN, whose output is given by:

$$y_n(x, \theta) = \sum_{i=1}^n A_{:i} \phi(B_{i:} x + b_i), \tag{1}$$

with $A \in \mathbb{R}^{l \times n}$, $B \in \mathbb{R}^{n \times d}$, $b \in \mathbb{R}^n$, and $\theta = \{A, B, b\}$. Note that here, and throughout the paper, we adopt the notation $X_{i:}$ and $X_{:j}$ to denote the $i^{th}$ row and $j^{th}$ column, respectively, of a matrix $X$. We shall investigate the approximation properties of this neural network when all the weights in $B$ and $A$ are fixed and sampled uniformly from the $n(l+d)$-dimensional centered hypercube, where the half-edges are of length $\gamma$ and only $b$ is tuned. We begin by outlining the activation function assumptions necessary for our theoretical results.

**Definition 1.** *The function $\phi$ is a* suitable activation *if, when employed in the neural network of Eq. 1, it allows for universal approximation of the following kind: for any continuous $h : U \to \mathbb{R}^l$ and any $\epsilon > 0$, $\exists n \in \mathbb{N}$ and parameters $\theta$ s.t. $\sup_{x \in U} ||h(x) - y_n(x, \theta)|| \le \epsilon$, where $U \subset \mathbb{R}^d$ is compact and $|| \cdot ||$ is the 1-norm and will be throughout the paper.*

From the universal approximation theorems of 1993 Leshno et al. (1993); Hornik (1993), a sufficient condition for $\phi$ to be a suitable activation is that it is non-polynomial. In this paper we conceptualize universal approximation as the approximation of continuous functions on compact sets with respect to an $L^\infty$ functional norm, but we remark that the literature has also studied other conditions on $h$ (for example measurability) and other forms of convergence. For a review of the literature, see Pinkus (1999).

**Definition 2.** *A suitable activation $\phi$ is referred to as a $\gamma$-parameter bounding activation if it allows for universal approximation even when each individual parameter, e.g. an element of a weight matrix or bias vector, is bounded by $\gamma$.*

**Proposition 1.** *The ReLU and the Heaviside step function are $\gamma$-parameter bounding activations for any $\gamma > 0$.*

The proof is in Appendix §B.2. A key subtlety of parameter-bounding is that it is a bound on *individual*, scalar, parameters. Thus, as a network grows in width the bias vector and weight matrix norms will still grow accordingly. This may be important, as research suggests band-limited parameters cannot universally approximate, at least for certain activation types Li et al. (2023). We leave it to future work to determine which other activations are parameter-bounding.

We make one final definition:

**Definition 3.** *If $\phi$ is a $\gamma$-parameter bounding activation, is continuous, and if $\exists \tau \in \mathbb{R}$ such that for $x < \tau\ \phi(x) = 0$, then we say that $\phi$ is a $\gamma$-bias-learning activation.*

Obviously, the ReLU is a bias-learning activation. We leave the study of discontinuous functions like the Heaviside to future work. We conclude with the main result of this section. Define $p_R$ to be a uniform distribution on $[-R, +R]$, where $\gamma < R < \infty$.

**Theorem 1.** *Assume that $\phi$ is $\gamma$-bias-learning and, for compact $U \subset \mathbb{R}^d$, $h : U \to \mathbb{R}^l$ is continuous. Then, for any degree of accuracy $\epsilon > 0$ and probability of error $\delta \in (0, 1)$, there exists a hidden-layer width $m \in \mathbb{N}$ and bias vector $b \in \mathbb{R}^m$ such that, with a probability of $1 - \delta$, a neural network given by Eq.1 with each individual weight sampled from $p_R$ approximates $h$ with error less than $\epsilon$.*

**Corollary 1.** *Assume that $d = l$, i.e. the input and output spaces of the network have the same dimension. Then the results of Theorem 1 also hold for single-hidden-layer ResNets.*

**Proof Intuition:** We provide intuition about the proof of Theorem 1 and its Corollary, whose details can be found in the Appendix (also see Fig. E.1 for visual proof intuition). According to the universal approximation theorem, given a continuous function, we can find a one-hidden-layer network, $\mathcal{N}_1$, that is close to that function in the $L^\infty$ norm on the (compact) space of its inputs. If $\mathcal{N}_1$ has been constructed using $\gamma$-parameter bounding activation functions, then we know that each parameter will be on the interval $[-\gamma, \gamma]$. Next, we construct a second network, $\mathcal{N}_2$, to approximate $\mathcal{N}_1$ by randomly sampling each of its parameters, weight or bias, from $p_R$. For $\mathcal{N}_2$ to approximate $\mathcal{N}_1$, each parameter of $\mathcal{N}_2$ should fall within a tiny window of an analogous parameter in $\mathcal{N}_1$. This window must have half-length less than $\epsilon$ to yield the desired error bound. Without loss of generality, we can assume $\epsilon < R - \gamma$. Then, if we sample parameters uniformly on $[-R, R]$, there will be a non-zero probability that a given parameter of $\mathcal{N}_2$ will end up within the tiny $\epsilon$-window centered at a corresponding parameter value in $\mathcal{N}_1$; because $\epsilon < R - \gamma$ we know that the $\epsilon$-window won't stretch outside the distribution support. If we randomly sample a *very* large number of units to construct the hidden layer of $\mathcal{N}_2$ the probability of finding a subnetwork of $\mathcal{N}_2$ corresponding to $\mathcal{N}_1$ can be made arbitrarily close to 1. If the activation function is bias-learning we can use biases to pick out this subnetwork by setting them appropriately smaller than the threshold given in Definition 3. We remark that this proof estimates exceedingly massive hidden-layer widths that, based on our numerical results, over-estimate the scaling of bias learning by orders of magnitude (see Remark 2 on Lemma 3). We thus view this proof not as a statement about scaling but as a statement of existence: for some sufficiently large but finite layer width, one can approximate the desired function with bias learning.

## 2.2 RECURRENT NEURAL NETWORKS

Here, we study a discrete-time RNN given by:

$$r_t = \alpha r_{t-1} + \beta \phi(W r_{t-1} + B x_{t-1} + b), \quad \hat{y}_t = C r_t, \tag{2}$$

where $r_t \in \mathbb{R}^m$ for all $0 \le t \le T$ for some $T \in \mathbb{N}$, $\alpha$ and $\beta$ control the time scale of the dynamics, $W \in \mathbb{R}^{m \times m}$, $C \in \mathbb{R}^{l \times m}$, and $B$ and $b$ are as in the previous section. The parameters are now $\theta = \{W, B, C, b\}$. The time-dependent input $x_t$ belongs to a compact subset $U_x \subset \mathbb{R}^d$ for all $t$. Note that when $\alpha = 0$, $\beta = 1$ one gets the standard vanilla RNN formulation; alternatively, $\alpha$ and $\beta$ can be set to approximate continuous-time dynamics using Euler's method.

We will approximate the following class of dynamical systems by learning only biases:

$$z_{t+1} = F(z_t, x_t), \quad y_t = Q z_t, \quad z_0 \in U_z, \tag{3}$$

where $t$ and $x_t$ are as defined for the RNN, $F : U_z \times U_x \to \mathbb{R}^s$ is continuous, and $Q \in \mathbb{R}^{l \times s}$. Because we build from the classic universal approximation results, we must be working with functions operating on compact sets. To guarantee that this will be the case we must make several more assumptions about the dynamical system. First, $U_z \subset \mathbb{R}^s$ is assumed to be a compact invariant set of the dynamical system: if the system is in $U_z$ it remains there for all $t$ and for all inputs in $U_x$. Second, we assume that the dynamical system is well-defined on a slightly larger compact set $\tilde{U}_z \times U_x$, where $\tilde{U}_z = \{z_0 + c_0 : z_0 \in U_z, ||c_0|| < c\}$ for some $c > 0$, with $\tilde{U}_z \supset U_z$.

To generalize the approximation in $L^\infty$ norm in our analysis of FNNs to dynamical systems we consider an *infinity norm over finite trajectories*: $\sup_{z_0 \in U_z, \mathbf{x}_t \in U_\mathbf{x}} \sum_{t=1}^{T} ||y_t(z_0, \mathbf{x}_t) - \hat{y}_t(r_0, \mathbf{x}_t)||$, where $\mathbf{x}_t \equiv [x_0, \ldots, x_{t-1}]$, $r_0 \equiv r_0(z_0)$ is a continuous map from $U_z$ into the hidden state space of the RNN, and $U_\mathbf{x}$ is the $t$-times product space of $U_x$. Letting $p_R$ be defined as in Section 2.1, the main result of this section is:

**Theorem 2.** *Consider the RNN in Eq.2 with $\phi$ a $\gamma$-bias-learning activation, and input, output, and recurrent weight parameters for each hidden unit sampled from $p_R$. We can find a hidden-layer width, $m$, a bias vector, and a continuous hidden-state initial condition map $r_0 : U_z \mapsto \mathbb{R}^m$ such that, with a probability that is arbitrarily close to 1, the RNN approximates the dynamical system defined in Eq.3 to below any positive, non-zero, error in the inifinity norm over trajectories.*

**Proof Intuition:** We provide a high-level description here while the detailed proof and a schematic (Fig. E.1) can be found in the Appendix. As in Theorem 1 the proof proceeds in two steps. First, the dynamical system is approximated by an RNN, $\mathcal{R}_1$, using universal approximation theory for RNNs (see e.g., Schäfer & Zimmermann (2006)). $\mathcal{R}_1$ is then approximated by a much wider, random RNN, $\mathcal{R}_2$, with parameters sampled from $p_R$. Analogous to Theorem 1, we show that one can find a sub-network of hidden units in $\mathcal{R}_2$ that approximates $\mathcal{R}_1$ for very large hidden widths of $\mathcal{R}_2$..

## 3 NUMERICAL RESULTS

### 3.1 MULTI-TASK LEARNING WITH BIAS-LEARNED FNNS

We first validated the theory by checking whether a single-hidden-layer bias-learned FNN could perform classification on the Fashion MNIST dataset Deng (2012) increasingly well as its hidden layer was widened. We compared fully-trained networks, matching the number of trained parameters, with bias-learned networks where the frozen weights were sampled from a uniform distribution on $[-0.1, 0.1]$ or a zero-mean Gaussian with standard deviation $\frac{1}{\sqrt{d}}$, where $d$ is the input dimension. The networks successfully learned the task and validation error decreased with the number of trained parameters (Fig. 1A) which, for bias-learned networks, is equal to the number of hidden units. The largest gains in performance occurred before $5 \times 10^3$ units, with bias-learning achieving comparable, but slightly worse, performance compared to fully-trained models. We speculate that this gap in performance for large network sizes is due either to current deep learning training conventions being optimized for weight training, or to bias learning requiring even larger hidden layer sizes to match fully-trained performance. Interestingly, for very low parameter counts bias-learning outperformed fully trained networks (more visible with log-log axis scaling E.2.A). Because weights scale quadratically with layer width, a fully-trained network will have a smaller width than a bias-learned network with the same number of trained parameters (see Fig.E.2.B for the error scaling plot with layer width on the $x$-axis).

Intuitively, bias learning should allow a single random set of weights to be used to learn multiple tasks by simply optimizing task-specific bias vectors. We confirmed this by training a single-hidden-layer FNN with $3.2 \times 10^4$ hidden units on 7 different tasks: MNIST Deng (2012), KMNIST Clanuwat et al. (2018), Fashion MNIST Xiao et al. (2017), Ethiopic-MNIST, Vai-MNIST, and Osmanya-MNIST from Afro-MNIST Wu et al. (2020), and Kannada-MNIST Prabhu (2019). All tasks involved classifying $28 \times 28$ grayscale images into 10 classes. The random weights were fixed across tasks while different biases were learned. We compared bias learning against a fully-trained neural network with the same size and architecture (Fig. 1B). We found that the bias-only network achieved similar performance to the fully-trained network on most tasks (only significantly worse on KMNIST). An important difference here is that the networks had matched size and architecture, so that the number of trainable parameters in the bias-only network ($3.2 \times 10^4$ parameters) was several orders of magnitude smaller than in the fully-trained case ($\approx 2.5 \times 10^7$ parameters). Notably, a

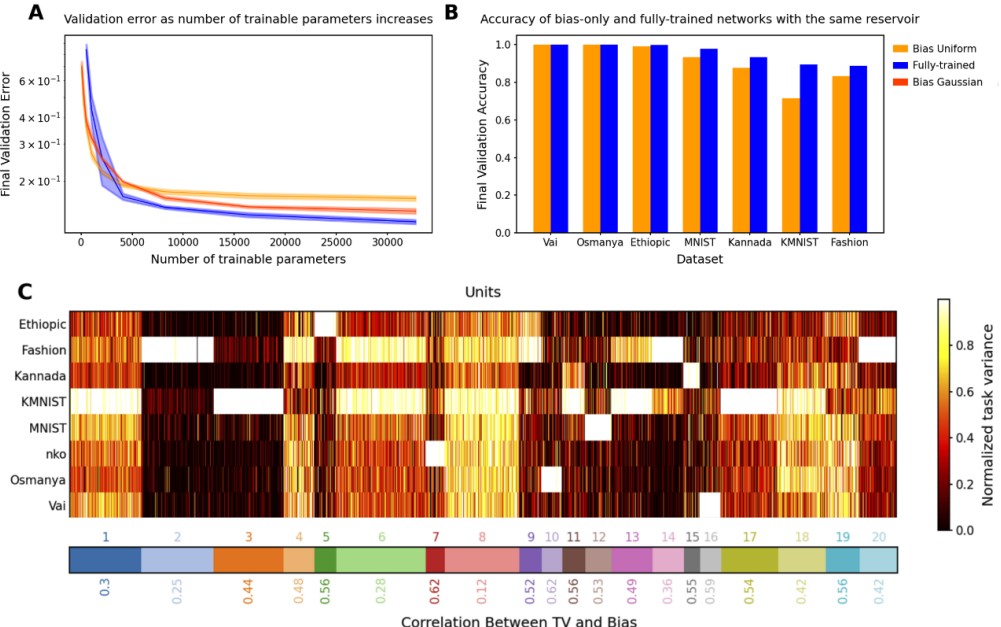

Figure 1: **A.** Validation accuracy on fashion MNIST vs. number of trained parameters for fully-trained (blue), bias-learned with uniformly distributed weights (light orange), and bias-learned with Gaussian weights (dark orange) networks. **B.** Validation accuracy on multiple image classification tasks for bias-learned (orange) and fully-trained (blue) networks. Errors for 5 random seeds are barely visible as the shaded regions in A, and are omitted in B because the standard errors are of order $10^{-3}$. **C.** Top: K-mean clustering of Task Variance (TV) reveals task-selective clusters (see Fig.E.3 for fully-trained network selectivity). Bottom: Spearman correlation between TV and bias vectors (mean across neurons in each cluster).

different set of biases was learned for each task. We conclude that bias-only learning in FNNs could be a viable avenue to perform multi-tasking with randomly initialized and fixed weights, but that it requires a much wider hidden layer than fully trained networks. Lastly, we note that the networks of Fig. 1B were trained with uniformly initialized weights, but that one can achieve similar, or even better performance with different weight initializations (see Fig. E.2C).

Next, we investigated the task-specific responses of hidden units by estimating single-unit Task Variances (TV) Yang et al. (2019), defined as the variance of a hidden unit activation across the test set for each task. The TV provides a measure of the extent that a given hidden unit contributes to the given task: a unit with high TV in one task and low TV for all others is seen as selective for its high-TV task. We clustered the hidden-unit TVs using K-means clustering ($K$ chosen by cross-validation) on the vectors of TVs for each unit and found that distinct clusters of units emerged (Fig. 1C). Some units reflected strong task selectivity (ex: cluster 3 for KMNIST and cluster 10 for Osmanya). Others responded to many, or all, tasks (ex: clusters 1 and 8), although a smaller fraction of clusters exhibited such non-selective activation patterns. Overall, we conclude that multi-task bias learning leads to the emergence of task-specific organization. We note, however, that task selectivity does not necessarily mean task utility: for example, a neuron could have a high variance for a single task but that variance could be picking up on noise and thus not functionally useful. We leave a deeper investigation of the functional significance of task-selectivity to future work.

Finally, we explored the relationship between the bias of a hidden unit and its TV. If the neural networks are using biases to shut-off units, analogous to the intuition in our theory (Section 2.1), then the units that do not actively participate in a task should be quiet due to a low bias value learned during training on that particular task. In other words, this intuition would suggest that units should exhibit a correlation between bias and TV, especially in task-specific clusters. In our experiments, all clusters did exhibit the statistical trend of a positive correlation between bias and TV, although to a varying degree across clusters (see numbers at the bottom of Fig. 1C).

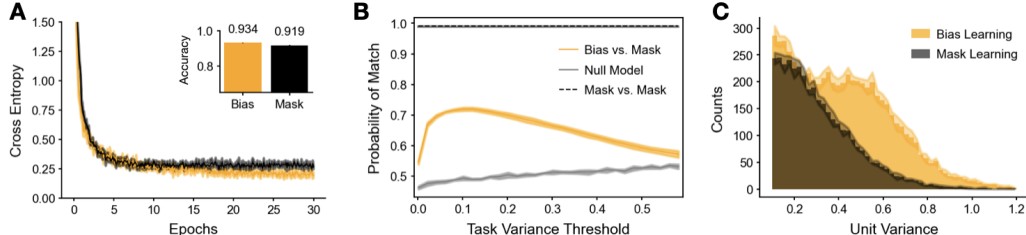

Figure 2: **Comparing bias and mask learning on same weights. A.** Learning curves for bias (orange) and mask (black) learning on MNIST. Inset: bias learning achieved roughly $1\%$ higher test accuracy over mask learning ($0.934 \pm 0.001$SD bias vs. $0.919 \pm 0.002$SD mask). **B.** Probability ($y$-axis) of the same unit being ON in both the bias-learning and mask-learning networks (orange line). A unit is 'ON' in mask learning if it is not masked out, and in bias learning if it has task variance above a given threshold ($x$-axis). Also shown is the probability of a unit being ON in two different training runs for mask-learning (black dashed line), and a null model giving the expected overlap if the probability of a unit being ON in the bias-trained network is independent of whether it is ON in the mask network (see Appendix §C.2 for more details) **C.** Histograms of hidden unit variances, calculated over $10^4$ test set MNIST samples, for bias-trained (orange) and mask-trained (black). Unit variances below $0.1$ are not shown. All curves, and histograms, are means, with shaded regions being 1SD over $5$ training runs.

## 3.2 RELATIONSHIP BETWEEN BIAS LEARNING AND MASK LEARNING IN FNNS

As our theory shows that bias learning networks can universally approximate simply by turning units off, we wished to test whether bias learning performs similarly to learning masks, and to what extent solutions learned by these approaches are different from each other. We compared training mask to bias learning on networks with the same random input/output weight matrices. For mask-training, we approximated binary masks using 'soft' sigmoid masks with learned gain parameters (see Methods). The approximation of a discontinuity with a differentiable function was done to allow optimization with gradient descent, a strategy with a history of use in ML Jang et al. (2016) and computational neuroscience Zenke & Ganguli (2018). The sigmoid was steepened over the course of training to approximate the binary mask that was used at test time. We compared masks learned in this fashion with learned biases on single-hidden-layer ReLU networks with $10^4$ units. We observed a trend of bias-training slightly improving upon mask-training (Fig.2A), which was expected given that biases can be tuned over a continuous range of values, including $0$ and $1$, while masks can only take $0$ or $1$. Further research is needed to determine if this trend is reliable across datasets and different network parametrizations, and whether there might be scenarios where one style of learning works better or worse.

Next, we compared the solutions found via bias and mask learning on the same set of randomly initialized weights. We calculated the variance of each hidden unit across $10^4$ MNIST test images, in both the bias and mask-trained paradigms, as a measure of hidden layer representation. To investigate whether the same units were active during the task regardless of training style, we looked at which units were 'ON' in mask versus bias-trained networks. For mask learning a unit was considered ON if its mask was 1; for bias learning a unit was ON if its task variance was above a given threshold. We observed that neurons with higher task variances contributed more to solving the task (Fig. E.4A). For a range of thresholds, we calculated the probability that a given unit was ON for both mask and bias learning (Fig.2B). We found that, for low thresholds, this match probability was intermediate between chance (grey line) and the high probability of a match between two mask-learned training runs on the same set of weights (black dashed line). We further noted that, on average, mask learning used $4527 \pm 44$ 'ON' units (in this section all reported values are mean$\pm$SD over $5$ samples) to solve the task. For bias learning, when the task variance threshold was moved from 0 to 0.1 test accuracy dropped about $1\%$ and the number of ON units went from $10^4$ (all units) to $6226 \pm 47$. Thus mask learning, found a sparser solution than bias learning. To further investigate the differences and similarities between unmasked units, we plotted the histograms of unit variances above the 0.1 threshold, and observed that bias learning used higher variances to accomplish the task (Fig.2C). Finally, we found a moderate correlation between the task variances of units that were

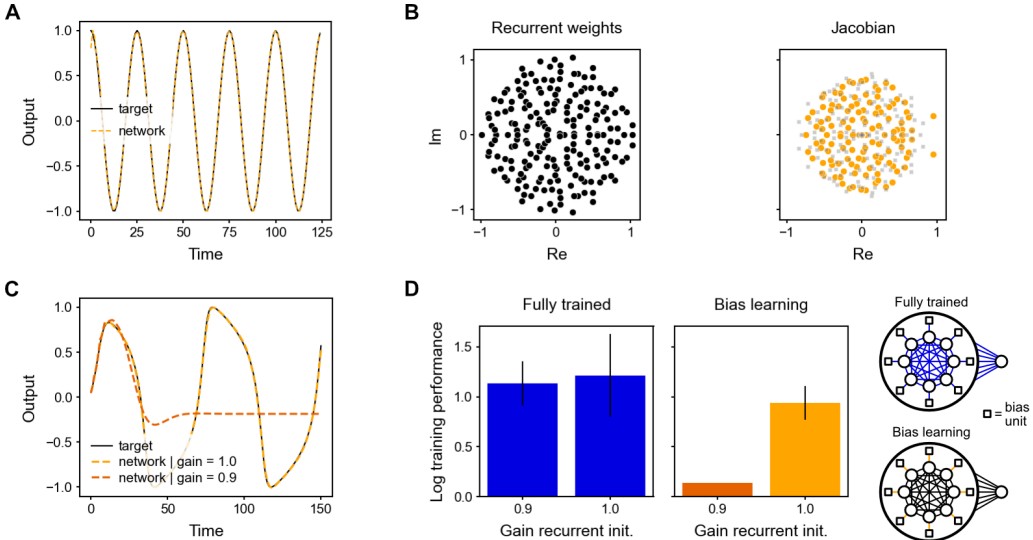

Figure 3: **Learning autonomous dynamical systems. A.** Cosine generated by a bias-learning RNN (dashed orange) and its target (solid black). **B.** Eigenvalue spectra for the recurrent weights (left) and the Jacobian at the start of training (right, grey squares) and mid-training (right, orange circles), when the network produced a decaying oscillation with period 23.75, close to the target period of 25. Neural activity then approached a fixed point with respect to which the Jacobian was computed. **C.** Van der Pol oscillator (target in solid black) generated by the bias-learning RNN for a recurrent gain of 1 (dashed orange; see panel D) and a gain of 0.9 (dashed dark orange). Output represents the oscillator's position, rescaled to [-1, 1]. **D.** (Left) Sensitivity to distribution of recurrent weights. The fully-trained and bias-learning networks had the same number of learnable parameters. Initial recurrent weight matrix had elements sampled from $(g/\sqrt{m})\mathcal{N}(0, 1)$, where $g$ is the gain (Gain recurrent init.). Error bars denote SEM for $n = 10$. (Right) Schematics of the fully-trained (top) and bias-learning (bottom) autonomous RNNs. Colored links denote trained weights.

ON for both mask and bias learning ($0.5030 \pm 0.0087$) (Fig E.4C). In summary, we observe that, relative to mask learning, bias learning finds a different, but overlapping, solution to MNIST.

## 3.3 BIAS LEARNING AUTONOMOUS DYNAMICAL SYSTEMS WITH RNNS

We studied the expressivity of bias learning in RNNs trained to generate linear and nonlinear dynamical systems autonomously (i.e., with $x_t \equiv 0$ in Eq. 2). We found that RNNs with fixed and random Gaussian weights and trained biases were able to generate a simple cosine function (Fig. 3A). We then elucidated the mechanism underlying RNN bias learning by comparing the Jacobian matrix after learning with the random recurrent weight matrix (which was held fixed during learning). We found that although the random weight matrix maintained a fixed and circular eigenvalue distribution (Fig. 3B, left), learning the biases shaped the Jacobian matrix to develop complex conjugate pairs of large eigenvalues underlying the oscillations (Fig. 3B, right). Therefore, bias learning strongly relies on the ability to shape the "effective connectivity matrix", i.e. the Jacobian, which involves the derivative of the activation and the recurrent weight matrix.

We next investigated whether bias learning relied on the statistics of the fixed recurrent weights. In light of Fig. 3B, we thus hypothesized that bias learning would be affected by changes in the weight distribution, because bias learning can only control the derivative. We initialized an i.i.d. Gaussian distributed weight matrix $W_{ij} \sim \mathcal{N}(0, g^2/N)$, where $g$ is referred to as its 'gain'. We then trained bias-learning networks to generate a van der Pol oscillator (Fig. 3C). We found that bias learning required a large enough gain (at least $g = 1$) and failed for $g < 1$ (Fig. 3D). This was not purely due to a restricted dynamic range for the network activity since the network was able to reproduce the first peak of the oscillator and then flatlined (Fig. 3C). In contrast, fully-trained networks with the same number of training parameters (Fig. 3D) were not sensitive to the value of

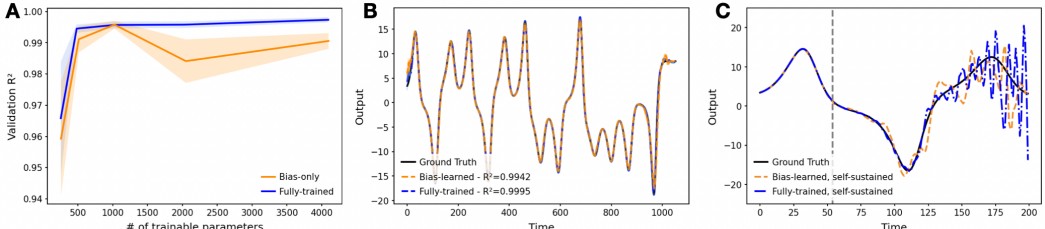

Figure 4: **Learning non-autonomous dynamical systems. A.** Validation $R^2$ vs. number of trainable model parameters for fully-trained (blue) and bias-learned (orange) RNNs. Training RNNs with bias learning became unstable below a network width of $64$. **B.** Predictions from the fully-trained and bias-learned networks (both with a hidden layer width of 1024) on a trajectory of the Lorenz system unseen during training. Standard deviation error bars were computed over 5 seeds, but are not visible due to their small magnitudes. **C.** Predictions of both the fully-trained and bias-learned networks diverge from the ground truth signal when one starts feeding back their own outputs as their inputs, in place of the ground-truth time-series (self-sustained, starting from the grey line).

the gain at initialization. This result thus highlights that, when the hidden-layer size is fixed, the initial distribution of weights limits the capability of bias-learning networks.

### 3.4 BIAS LEARNING NON-AUTONOMOUS DYNAMICAL SYSTEMS WITH RNNS

To further test bias learning in RNNs, we trained a RNN to predict future time-steps of a partially observed dynamical system, namely a single dimension of the Lorenz attractor (see Appendix §C.4 for details). As in the autonomous DS case, only the biases of the input layer were trained and the weights were random and frozen. However, here the network received the observed dimension of the Lorenz system as an input. Given the observed dimension at time-point $t$, and its value at previous time-steps encoded in the RNN's hidden state, the objective of the task was to predict the future value at $t + \tau$, where $\tau = 27$ was chosen to be the half-width at the half-max of the auto-correlation of the observed dimension of the Lorenz system.

The performance of the bias-learned RNNs scaled, as a function of trainable model parameters, in a qualitatively similar fashion to the bias-trained FNNs (Fig. 4A; see Fig. E.2D for scaling as function of layer width). Both the fully-trained and sufficiently wide bias-learned networks accurately predicted future points of the Lorenz system, evidenced by a consistent $R^2$ metric of $> 0.99$ ($n$=5) achieved by networks with a hidden-layer width of 1024 on a window of the Lorenz time-series held out during training (Fig. 4B). However, when the networks were fed their own previous predictions as input, in place of the ground-truth time series, in an autoregressive, 'self-sustained' fashion, their prediction accuracy decreased, demonstrating the damaging effect of small compounding deviations propagated through time (Fig. 4C).

### 3.5 BIAS-LEARNED MOTOR CONTROL WITH RNNS

Finally, we tested whether bias learning could solve a center-out reaching task, a paradigm routinely used to study motor control in human and non-human primates Ashe & Georgopoulos (1994); Shadmehr & Mussa-Ivaldi (1994). Starting from the center of the workspace, the subject must move the selected end effector (e.g., their right hand) to reach several peripheral targets placed equidistantly on a circle. We modelled this task using an RNN with random weights and learned biases, where linear readouts controlled a point-mass arm (a unit mass modeling the arm's behavior; see Methods). The task objective was to reach the peripheral targets in 1 second, with near-zero velocity and force at the end of movement, which were imposed using regularization terms in the loss function. A 1,024-units network required approximately $10^4$ training epochs—where one epoch involved presenting all targets once—to solve the task (Fig. 5A). (Note that a parameter-matched, fully-trained RNN can also solve this task, in $\sim 10^3$ epochs (Fig. E.5).) Importantly, a single set of biases was used to reach all 6 targets for each trained network. The resulting trajectories successfully reached the targets (Fig. 5B, dark curves and black targets), and exhibited the characteristic bell-shaped velocity profile Harris & Wolpert (1998) (Fig. 5C, top). Crucially, the trained network generalized

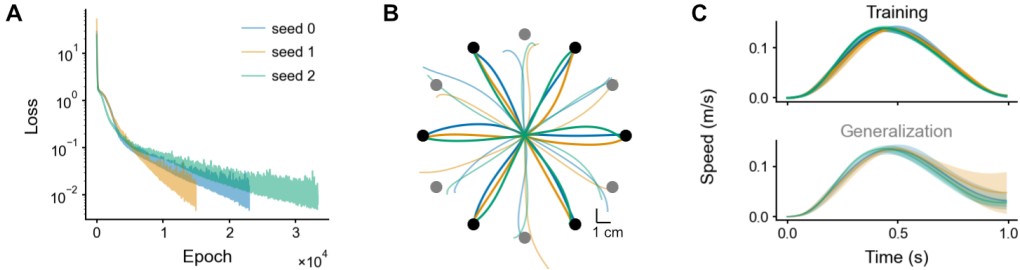

Figure 5: **Center-out reaching task. A.** Training loss for 3 network initializations. **B.** Trajectories for the trained (black) and tested (grey) targets. **C.** Speeds ($(\dot{x}^2 + \dot{y}^2)^{1/2}$) for the trained (top) and tested (bottom) targets (mean $\pm$ SD across targets).

decently to new targets on the circle (Fig. 5B, light curves and grey targets; Fig. 5C, bottom). To achieve this, the network had to produce both acceleration and deceleration when given information about the Cartesian position of a target never seen in the training period. These results highlight the flexibility of bias learning in generating diverse open-loop controls.

## 4 DISCUSSION

In this paper, we presented theoretical results demonstrating that FNNs and RNNs with fixed random weights but learnable biases can approximate arbitrary functions with high probability. We showcased the expressivity of bias-learned networks in multi-task, times-series, and motor control tasks, and interrogated their learned representations by analyzing task selectivity and comparing with mask-learning in FNNs, and by analyzing eigenvalues in RNNs.

We underline four limitations of our study that might inspire future research. First, the convergence results for dynamical systems were only for finite-time trajectories. One could overcome this limitation by studying convergence in stationary distribution. Second, a potential confounding factor in our comparison of bias and mask learning is that the latter approach used a learning schedule in the steepness parameter for the soft masks. It is possible that the altered learning dynamics due to this scheduling contributed to mask and bias learning finding different solutions. Addressing this confound is an important direction for future work. Third, a better grasp of bias learning scaling–how layer width or parameter count increases as a function of performance–is needed. Our theory does not shed light here as it significantly overestimates how a bias-learned network should scale (Remark 2 on Lemma 3 in Appendix). Some insight can be gleaned by noting that, with bias-learning activations, tuned biases can express any mask-learned solution. Thus, results showing that mask learning layer widths scale polynomially in the inverse error and the size of a performance-matched random feature model (see Malach et al. (2020) Theorem 3.2) represent a worst-case scaling for bias learning. Whether bias learning scales better than mask learning could be a good starting point for addressing scaling. Fourth, while our results appeared to generalize beyond one-layer networks (Fig. E.2E), a more detailed study of bias-learned deep FNNs is left to future work.

We lastly highlight the need for greater biological detail in future studies of bias learning. Past experimental Ferguson & Cardin (2020) and theoretical work Wyrick & Mazzucato (2021); Ogawa et al. (2023) showed that neural mechanisms modulating biases, like firing threshold or tonic inputs, may effect other neuronal properties, like neuron input-output gain. As our proofs rely on masking, they demonstrate universal approximation not just for bias learning but for any learned mechanism that can mask neurons. Exploring paradigms where gain (Stroud et al. (2018)) and biases are learned in concert could be an interesting direction to exploit this. Finally, the observed distribution of synaptic weights in the brain is not uniform but long-tailed Song et al. (2005), highlighting the need for bias learning models with more structured weight initializations. We hypothesize that structure in the weights intermediate between fully random and fully learned might yield an optimal combination of performance and training efficiency. If such structure improves hidden-layer scaling, this could enable bias learning to support temporal credit assignment algorithms that struggle in weight space due to the $N^2$ scaling of synapses. A fascinating alternative is that the same number of parameters could achieve a given task performance, regardless of whether they are weights or biases.

ACKNOWLEDGEMENTS

E.W. was supported by an NSERC CGS D scholarship, and wishes to thank the other members of the Lajoie lab for support and for helpful discussions. T.J. was supported by an NSERC CGS M scholarship. M.G.P. was supported by grant the Fonds de recherche du Québec – Santé (chercheurs-boursiers en intelligence artificielle). L.M. was partially supported by National Institutes of Health grants R01NS118461, R01MH127375 and R01DA055439 and National Science Foundation CA-REER Award 2238247. GL acknowledges CIFAR and Canada chair program.

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

## A    BIOLOGICAL BIAS-RELATED MECHANISMS

| System | Bias model | Mechanism | Effect | Refs |
|---|---|---|---|---|
| Working memory | Dopamine projections to cortical circuit | Bias of NMDA or GABA conductances | Enhance memory signal-to-noise ratio and resistance to distractors | Brunel & Wang (2001) |
| Behavioral modulation of visual responses | Thalamic projections to visual cortex | Bias to pyramidal and somatostatin cells | Paradoxical reversal of somatostatin cell responses to running | Garcia del Molino et al. (2017) |
| Motor sequences | Basal ganglia projections to thalamus | Inhibitory biases of thalamic neurons | Toggle on/off different motor primitives | Logiaco et al. (2021) |
| Motor sequences | Secondary motor cortex projections to primary motor cortex (M1) | Biases of M1 neurons | Select initial conditions (anticipatory activity) for motor primitives | Recanatesi et al. (2022); Mazzucato (2022) |
| Short-term motor adaptation | Premotor cortex projections to primary motor cortex (M1) | Recruitment of M1 neurons | Shift pre-movement preparatory states | Perich et al. (2018); Feulner et al. (2022) |
| Expectation and taste processing | Amygdalar projections to gustatory cortex | Biases of cortical neurons | Acceleration of taste coding via gain modulation | Mazzucato et al. (2019); Vincis & Fontanini (2016); Haley et al. (2020) |
| Movements and visual processing | Thalamic projections to visual cortex | Biases of cortical neurons | Acceleration of visual coding via gain modulation | Wyrick & Mazzucato (2021) |
| Arousal and auditory processing | Neuromodulatory projections to auditory cortex | Biases of cortical neurons | Optimal encoding of auditory stimuli at intermediate arousal | Papadopoulos et al. (2024) |

Table 1: **Potential biological mechanisms for bias-related plasticity in the brain.**

## B    MATHEMATICAL PROOFS

Throughout the appendix the proofs are restated for ease of reference. We will always take $||\cdot||$ to be the 1-norm unless stated otherwise.

### B.1    RANDOM NEURAL NETWORK FORMULATION

The proofs of this section revolve around masked, random, neural networks:

$$\tilde{r}_m^{\mathcal{M}} = \alpha r_0 + \mathcal{M} \odot \beta\phi(Wr_0 + Bx + b), \quad \tilde{y}_m^{\mathcal{M}} = A\tilde{r}_m^{\mathcal{M}}, \tag{4}$$

where $\alpha \in \mathbb{R}$, $0 < \beta < \infty$, $m \in \mathbb{N}$, $r_0$, $\tilde{r}_m^{\mathcal{M}} \in \mathbb{R}^m$, $x \in \mathbb{R}^d$, $\tilde{y}_m^{\mathcal{M}} \in \mathbb{R}^l$, $\mathcal{M} \in \{0,1\}^m$, and all other matrices and vectors have real elements with the dimensions required by the above definitions. We assume that $\phi$ is $\gamma$-parameter bounding and that each individual (scalar) parameter, be it weight or bias, is sampled randomly–before masking–from a uniform distribution on $[-\bar{\gamma}, \bar{\gamma}]$ (note that here we are using $\bar{\gamma}$ where we used $R$ in the main text). In this way the parameters are random variables with compact support. If $\mathcal{M} = \mathbf{1}$ then we drop the superscript. To account for feed-forward neural networks we simply set $W$ to the zero matrix and $\alpha, \beta = 0, 1$ (this is assumed in Section B.2).

W.l.o.g. assume there are $n$ non-zero elements in $\mathcal{M}$. We construct $W^{\mathcal{M}} \in \mathbb{R}^{n \times n}$–the recurrent matrix restricted to participating (non-masked) hidden units–by beginning with $W$ and deleting the $i^{th}$ row and $i^{th}$ column of the matrix if $\mathcal{M}_i = 0$. We construct $B^{\mathcal{M}} \in \mathbb{R}^{n \times d}$, $A^{\mathcal{M}} \in \mathbb{R}^{l \times n}$, and $b^{\mathcal{M}} \in \mathbb{R}^n$ by deleting the $i^{th}$ row of $B$, $A$, and $i^{th}$ element of $b$ if $\mathcal{M}_i = 0$.

Consider the case where the $i^{th}$ element of $r_0$ is 0 whenever $\mathcal{M}_i = 0$. Then, regardless of whether Eq.4 represents a feed-forward network or the transition function for an RNN, the masked units will always be zero. We can thus simply track the $n$ units that correspond with 1's in $\mathcal{M}$ as the outputs, $y^{\mathcal{M}}$ will depend solely on these. We observe that the behaviour of these units can be described by the following network:

$$r_m^{\mathcal{M}} = \alpha r_0 + \beta\phi(W^{\mathcal{M}}r_0 + B^{\mathcal{M}}x + b^{\mathcal{M}}), \quad y_m^{\mathcal{M}} = A^{\mathcal{M}}r_m^{\mathcal{M}}. \tag{5}$$

It is networks of the form of Eq.5 that will be the primary subject of study in what follows. Note that the '$\sim$', over the $r$, is dropped to denote the fact that $r$ is a different vector on account of dropping the zero units. In the feed-forward case we use subscripts, as we have done above, to denote hidden layer width. Whenever we discuss RNNs or dynamical systems we will instead use the subscript to denote time.

## B.2 Proofs from Section 2.1

**Proposition 1.** *The ReLU and the Heaviside step function are $\gamma$-parameter bounding activations for any $\gamma > 0$.*

*Proof.* We prove this solely for the ReLU, as the logic for the Heaviside is effectively the same. Let $\phi$ thus be a ReLU. First, observe the following useful property: for all $a > 0$ we have $a\phi(x) = \phi(ax)$. From this, consider the neural network of hidden layer width $n$ with ReLU activations, $y_n(\theta)$, and observe:

$$y_n(\theta) = a^2 \sum_{i=1}^{n} \frac{A_{:i}}{a} \phi\left(\frac{B_{i:}}{a}x + \frac{b_i}{a}\right) = a^2 y_n\left(\frac{\theta}{a}\right). \tag{6}$$

Moreover, if $a \in \mathbb{N}$ we have

$$y_n(\theta) = \sum_{i=1}^{a^2 n} \tilde{A}_{:i}\phi\left(\tilde{B}_{i:}x + \tilde{b}_i\right) = y_{a^2 n}(\tilde{\theta}), \tag{7}$$

where $\tilde{\theta} = [\frac{\theta_1}{a}, \ldots, \frac{\theta_n}{a}, \ldots \frac{\theta_1}{a}, \ldots, \frac{\theta_n}{a}]$ so that each element is simply a re-scaled and repeated version of the original parameters; we have $a^2$ repeats for each term to replace the $a^2$ factor in the LHS of Eq. 6.

Now, given an arbitrary compact set $U \in \mathbb{R}^d$, continuous function $h : \mathbb{R}^d \to \mathbb{R}^l$, and $\varepsilon > 0$, by the universal approximator theory (see e.g. Leshno et al. (1993); Hornik (1993)) we can find $n$ such that

$$\sup_{x \in U} ||h(x) - y_n(x, \theta)|| \leq \epsilon \tag{8}$$

holds. Because $n$ is finite we can bound every individual (scalar) parameter by $M$, for some sufficiently large $M$. Suppose we want the parameters to be bounded instead by $\gamma$ with $M > \gamma > 0$. If we select $a \in \mathbb{N}$ s.t. $a > \frac{M}{\gamma}$ then we can find $y_{a^2 n}(x, \tilde{\theta})$ such that $y_{a^2 n}(x, \tilde{\theta}) = y_n(x, \theta)$. Thus we have found a parameter-bounding ReLU neural network satisfying Eq.8, completing the proof.

$\square$

**Remark:** The intuition behind this result, for the ReLU, is credited to a reply to the Universal Approximation Theorem with Bounded Parameters question on Mathematics Stack Exchange.

The following lemma constitutes the core of Theorem 1. It shows that one can achieve universal approximation, in the sense needed for the theorem, using masking. The theorem then follows by manipulating biases to achieve masking.

**Lemma 1.** *Let $h : U \to \mathbb{R}^l$ be a continuous function on compact support $U \subset \mathbb{R}^d$. Then for any $\epsilon > 0, \delta \in (0, 1)$, we can find a layer width $m \in \mathbb{N}$ such that with probability at least $1 - \delta$ $\exists \mathcal{M} \in \{0, 1\}^m$ satisfying the following:*

$$\sup_{x \in U} ||h(x) - y_m^{\mathcal{M}}(x)|| \leq \epsilon. \tag{9}$$

*Proof.* First, we find a neural network with parameters that approximate the desired function $h$. Given the assumptions on $\phi$, we can use Proposition 2 to find $n$ and parameters $\theta^* = \{A^*, B^*, b^*\}$ such that

$$\sup_{x \in U} ||h(x) - y_n(x, \theta^*)|| \leq \frac{\epsilon}{2}, \tag{10}$$

because $U$ is compact and $h$ is continuous.

We now make two observations: first, all choices of $x$ are from a compact set, $U$, by assumption and the parameters of a given random or non-random neural net are also from a compact set–the $n$ dimensional hyper-cube with edge length $2\bar{\gamma}$. Second, the function $y_n$ is a continuous function of $x$ and the parameters. By these two observations the function $y_n$ is a continuous function on the compact product space of inputs and parameters, and thus admits a Lipshitz constant, $K_n$. This will come in handy momentarily.

Next, we construct a masked random network that approximates $y_n$ with high probability. By Lemma 3, we can find a random feed-forward neural network of hidden layer width $m$ such that a mask, $\mathcal{M}$, exists satisfying $|\theta_i^* - \theta_i^{\mathcal{M}}| < \varepsilon$ for some arbitrarily $\varepsilon > 0$. In particular, we can choose $\varepsilon$ as:

$$|\theta_i^* - \theta_i^{\mathcal{M}}| < \varepsilon = \frac{\epsilon}{2 K_n n (d + l + 1)} \tag{11}$$

for all $i$ with probability at least $1 - \delta$. If we are in the regime of probability $1 - \delta$ where the mask satisfying the above error bound exists then we get

$$||y_m^{\mathcal{M}}(x) - y_n(x, \theta^*)|| \leq K_n \big(||\theta^{\mathcal{M}} - \theta^*|| + ||x - x||\big) \leq K_n n (d + l + 1) \varepsilon \leq \frac{\epsilon}{2}, \tag{12}$$

where, in addition to Eq.11, we used the fact that $y_m^{\mathcal{M}}(x) = y_n(x, \theta^{\mathcal{M}})$, the continuity and compactness mentioned above, and repeated application of the triangle inequality. Importantly, this bound holds for all $x \in U$. Because $\phi$ is assumed continuous, the function $f(x) = ||y_m^{\mathcal{M}}(x) - y_n(x)||$ is also continuous. By the extreme value theorem $\exists\, x^\star \in U$ such that $\sup_{x \in U} f(x) = f(x^\star)$. Since $x^\star \in U$ the bound from Eq.12 applies and we have:

$$\sup_{x \in U} ||y_m^{\mathcal{M}}(x) - y_n(x)|| = ||y_m^{\mathcal{M}}(x^\star) - y_n(x^\star)|| \leq \frac{\epsilon}{2}. \tag{13}$$

Using the triangle inequality, Eq.10, and Eq.13 gives $\sup_{x \in U} ||h(x) - y_m^{\mathcal{M}}(x)|| \leq \epsilon$ with probability $1 - \delta$.

$\square$

**Connections with Malach et al. 2020:** As mentioned in the main text, the previous lemma is closely related to past results on the SLTH over units for MLPs with one hidden layer. In Theorem 3.2 of Malach et al. (2020), it is proven that one can match the performance of a random feature model (an MLP where only the output layer is trained) by scaling the output of a mask-learned network. This theorem differs from ours in three important ways. First, it proves a result for a potentially different class of activation functions, second, it requires a scaling of the output of the network–unlike our proof which does not require this–and, third, it compares the performance of mask-learned networks to random feature models, rather than directly proving that mask-learned networks can approximate wide classes of functions. We believe that one should be able to achieve a result similar to ours by combining Malach et al.'s Theorem 3.2 with Theorem 1 of Rahimi & Recht (2008) (or a similar result on learning with random networks), but we leave the details of this to future studies.

**Theorem 1.** *Assume that $\phi$ is $\gamma$-bias-learning and, for compact $U \subset \mathbb{R}^d$, $h : U \to \mathbb{R}^l$ is continuous. Then, for any degree of accuracy $\epsilon > 0$ and probability of error $\delta \in (0, 1)$, there exists a hidden-layer width $m \in \mathbb{N}$ and bias vector $b \in \mathbb{R}^m$ such that, with a probability of $1 - \delta$, a neural network given by Eq.1 with each individual weight sampled from $p_R$ approximates $h$ with error less than $\epsilon$.*

*Proof.* Observe that, once we have choosen an $m$ satisfying the desiderata of Lemma 1, because $\phi$ is assumed to be $\gamma$-bias-learning, $m$ is some finite value and all variables that make up the input of $\phi$ are bounded, we can implement the mask by setting $b_i$ to be very negative for every $i$ such that $\mathcal{M}_i = 0$. For every $b_i$ such that $\mathcal{M} = 1$ we simply leave $b_i$ at its original randomly chosen value. $\square$

**Corollary 2.** *Assume $d = l$, that is, the output and input spaces are the same. Then the results of Lemma 1 and Theorem 1 also hold for res-nets; that is, networks whose output is of the form $x + y_m^{\mathcal{M}}(x)$.*

*Proof.* This follows by observing that $h(x) + x$ is also a continuous function and then replacing $h(x)$ with $h(x) + x$ in Eq.9 and rearranging. $\square$

**Remark:** While the error can be made arbitrarily small, the limit of zero error itself is undefined. This is because our proof relies on first approximating the given smooth function with a neural network with all parameters tuned and then approximating this second network using bias-learning to pick-out a matching sub-network from a large random reservoir; the probability of perfectly matching the fully tuned network with the bias-learned network is zero. This could be addressed by using an integral representation for continuous functions instead of directly using a finite-width neural network to approximate the given function (see e.g. Rahimi & Recht (2008); Li et al. (2023)). As one will see below, this remark also applies to the recurrent neural network result.

### B.3 PROOF FROM SECTION 2.2

Analogous to the section containing the feed-forward proofs, we first state and prove a lemma which comprises the core of the proof for recurrent neural networks. This lemma shows that one can achieve universal approximation in the $L^\infty$ norm over trajectories sense (see section §2.2) with high probability using masking in a randomly initialized RNN, and in this way provides a proof of the SLTH over units for RNNs. The proof of the main theorem in this section then follows quite straightforwardly.

**Lemma 2.** *Consider a discrete time, partially observed dynamical system of the form of, and satisfying the same conditions as, the one in Eq.3. Let $0 < T < \infty$, $x_t \in U_x \ \forall t$, $\epsilon > 0$ and $\delta \in (0, 1)$. Then we can find an RNN with appropriately chosen initial conditions and a layer width $m \in \mathbb{N}$ such that with probability at least $1 - \delta \ \exists \mathcal{M} \in \{0, 1\}^m$ satisfying the following:*

$$\sup_{z_0 \in U_r, x_{0:(T-1)} \in U_{\bar{x}}} \sum_{t=1}^{T} ||y_t - y_t^{\mathcal{M}}|| < \epsilon. \tag{14}$$

*Proof.* It is well known that we can arbitrarily approximate this dynamical system with an RNN Schäfer & Zimmermann (2006); we provide a simple proof of this in Proposition 3. In particular, for arbitrary $\epsilon > 0$ we can find an RNN of the form in Eq.2, with hidden layer width $n \in \mathbb{N}$ and output $\hat{y}$, satisfying:

$$\sup_{z_0 \in U_r, x_{0:(T-1)} \in U_{\vec{x}}} \sum_{t=1}^{T} ||\hat{y}_t - y_t|| < \frac{\epsilon}{2}, \tag{15}$$

Consider an $n$-width RNN of the form of Eq.2 with parameter-bounding activation functions and with initial conditions $r_0(z_0)$, for $z_0 \in U_z$, determined by Eq.30. This will result in $y_t = y_t(z_0, x_{0:(t-1)}, \theta)$ being a continuous function in all of its arguments. Moreover, all of its arguments are from compact sets: $U_z$ is compact, $U_{\vec{x}}$ is compact, and $\theta$ can be chosen to be on the hypercube with half-edges of length $\bar{\gamma}$. Thus $y_t$ admits a Lipschitz constant $K_n$. We make use of this below.

Let the parameters of the above-defined approximating RNN, from Eq.15, be given by $\theta^* = \{A^*, W^*, B^*, b^*\}$. Then by Lemma 3 we can find a random RNN of hidden width $m$ and with parameters $\theta$ such that a mask, $\mathcal{M}$, exists satisfying

$$|\theta_i^* - \theta_i^{\mathcal{M}}| < \varepsilon = \frac{\epsilon}{2K_n T n(n+l+d+1)}, \tag{16}$$

for all $i$ with probability at least $1 - \delta$. Then, by the Lipshitz property of $y_t$, we get

$$\sum_{t=1}^{T} ||y_t^{\mathcal{M}} - \hat{y}_t|| \leq \sum_{t=1}^{T} K_n \left( ||\theta^{\mathcal{M}} - \theta^*|| + ||x_{0:(t-1)} - x_{0:(t-1)}|| + ||z_0 - z_0|| \right) \tag{17}$$

$$\leq K_n T n(n+l+d+1)\varepsilon. \tag{18}$$

This follows from $y_t^{\mathcal{M}} = y_t(r_0^{\mathcal{M}}, x_{0:(t-1)}, \theta^{\mathcal{M}})$–where we have defined $r_0^{\mathcal{M}}$ to be equal to $r_{0i}$ for all $i$ such that $\mathcal{M}_i = 1$ and 0 for all other indices–and the fact that both $r_0$ and $r_0^{\mathcal{M}}$ are continuous functions of $z_0$. This bound is independent of both inputs and initial condition, and, for any parameters within the hypercube defined by $\bar{\gamma}$, holds for all $z_0 \in U_z$ and inputs $x_{0:(T-1)} \in U_{\vec{x}}$. By the continuity assumptions on the activation function, $f(z_0, x_{0:(T-1)}) \equiv \sum_{t=1}^{T} ||y_t^{\mathcal{M}} - \hat{y}_t||$ is a continuous function of $z_0$ and $x_{0:(T-1)}$ for any $\theta^*$ and $\theta^{\mathcal{M}}$ meaning that, analogous to the feed-forward case studied previously, we can use the extreme value theorem to find maximal values $z_0^\star \in U_z$ and $x_{0:(t-1)}^\star \in U_{\vec{x}}$ such that $\sup_{z_0 \in U_z, x_{0:(t-1)} \in U_{\vec{x}}} f(z_0, x_{0:(t-1)}) = f(z_0^\star, x_{0:(t-1)}^\star)$. It thus follows that $\sup_{z_0 \in U_r, x_{0:(T-1)} \in U_{\vec{x}}} \sum_{t=1}^{T} ||y^{\mathcal{M}} - \hat{y}_t|| < \frac{\epsilon}{2}$, by Eq's 16 and 18. The triangle inequality along with Eq.15 completes the proof.

$\square$

**Theorem 2.** *Consider the RNN in Eq.2 with $\phi$ a $\gamma$-bias-learning activation, and input, output, and recurrent weight parameters for each hidden unit sampled from $p_R$. We can find a hidden-layer width, $m$, a bias vector, and a continuous hidden-state initial condition map $r_0 : U_z \mapsto \mathbb{R}^m$ such that, with a probability that is arbitrarily close to 1, the RNN approximates the dynamical system defined in Eq.3 to below any positive, non-zero, error in the inifinity norm over trajectories.*

*Proof.* This follows directly from Lemma 2, by observing that one can replace the mask by simply setting biases to some sufficiently low value. $\square$

### B.4 SUPPLEMENTARY LEMMAS

The following result is well known in the literature; see e.g. Proposition 1 of Leshno et al. (1993).

**Proposition 2.** *For any $\epsilon > 0$ $\exists n \in \mathbb{N}$ s.t.*

$$\sup_{x \in U} ||h(x) - y_n(x, \theta^*)|| \leq \epsilon \tag{19}$$

**Corollary 3.** *The above holds if we restrict the output weight matrix of the neural network to have rank equal to the output dimension.*

*Proof.* This is because the set of full rank matrices is dense in $\mathbb{R}^{m \times n}$ for $m, n \in \mathbb{N}$. $\qquad\square$

Consider matrices $W^* \in \mathbb{R}^{n \times n}$, $B^* \in \mathbb{R}^{n \times d}$, $A^* \in \mathbb{R}^{l \times n}$, and vector $b^* \in \mathbb{R}^n$. We can vectorize and concatenate their elements into the single long vector $\theta \in \mathbb{R}^\pi$, where $\pi = n(n + d + l + 1)$. Assume that $|\theta_i^*| < \gamma$ for all $i$.

Next, construct $W \in \mathbb{R}^{m \times m}$, $B \in \mathbb{R}^{m \times d}$, $A \in \mathbb{R}^{l \times m}$, and vector $b \in \mathbb{R}^m$, by sampling each element randomly from a uniform distribution on $[-\bar{\gamma}, \bar{\gamma}]$ where $\bar{\gamma} = \gamma + \Delta\gamma$ for $\Delta\gamma > 0$. We analogously group these into a single vector, $\theta \in \mathbb{R}^{m(m+d+l+1)}$ Observe that for each $\mathcal{M} \in \{0,1\}^m$, we can construct sub-matrices of $W$, $B$, $A$, and sub-vector of $b$ by deleting column and row pairs in $W$, rows in $B$, columns in $A$, and elements of $b$ whose indices correspond to $i \in \{1, \ldots, m\}$ such that $\mathcal{M}_i = 0$. For a given $\mathcal{M}$, we define $\theta^\mathcal{M}$ to be the vector constructed by flattening and concatenating these sub-matrices and vector. We then have the following lemma:

**Lemma 3.** *For $\theta^*$, defined above, and arbitrary $\varepsilon > 0$, $\delta \in (0, 1)$, we can find $m > n$ such that with probability at least $1 - \delta$ $\exists \mathcal{M} \in \{0, 1\}^m$ with only $n$ non-zero elements such that $|\theta_i^* - \theta_i^\mathcal{M}| < \varepsilon$ for all $i \in \{1, \ldots, \pi\}$. In particular, any $m \geq \frac{n \log \delta}{\log[1 - (\frac{\epsilon}{\bar{\gamma}})^\pi]}$ will satisfy the result, where $\epsilon = \min(\varepsilon, \Delta\gamma)$.*

*Proof.* In what follows we set $\epsilon = \min(\varepsilon, \Delta\gamma)$. This simplifies the below probability bound that we derive because it means the probability of falling within an $\epsilon$ window of a desired parameter will not change, even if the desired parameter is very close to its bound, $\pm\gamma$. We will refer to the event that the desiderata of the lemma are satisfied for $\epsilon$, rather than $\varepsilon$, as $A_1$; that is: $\exists \mathcal{M} \in \{0, 1\}^m$ with only $n$ non-zero elements such that $|\theta_i^* - \theta_i^\mathcal{M}| < \epsilon$ for all $i \in \{1, \ldots, n\}$. The event that the desiderata are not satisfied is $A_1^c$.

Assume that $m^* = kn$ for some $k \in \mathbb{N}^+$. Consider the *'block' mask* $\mathcal{M}^{k_1}$ s.t. $\mathcal{M}_i^{k_1} = 1$ only for $i \in \{(k_1 - 1)n + 1, \ldots, k_1 n\}$, with $0 < k_1 \leq k$. Note that the $n$ elements selected by these block masks are non-overlapping for two different $k_1$. Let event $A_2$ be the event that there is a block mask that occurs satisfying the desiderata of the lemma with error $\epsilon$. Clearly $A_2 \subset A_1 \implies A_1^c \subset A_2^c \implies P(A_1^c) \leq P(A_2^c)$. $A_2^c$ is the probability that there is no block mask satisfying the desiderata. Observe that

$$P(A_2^c) = P\left[ \bigcap_{k_1=1}^k \{k_1^{th} \text{ block mask doesn't work}\} \right] = \prod_{k_1=1}^k P(\{k_1^{th} \text{ block mask doesn't work}\})$$

$$= \prod_{k_1=1}^k 1 - P(\{k_1^{th} \text{ block mask works}\}) = \left[ 1 - \left( \frac{\epsilon}{\bar{\gamma}} \right)^\pi \right]^{\frac{m^*}{n}}, \tag{20}$$

which follows from the fact that the elements of the matrices are independently sampled and the elements corresponding to sub-matrices selected by a given block mask are independent of those associated with another block mask. By making $m^*$ *very* large we can make $P(A_2^c)$ arbitrarily small. Because $P(A_1^c) \leq P(A_2^c)$–and the desiderata of the lemma with error $\epsilon$ are not satisfied solely on $A_1^c$–the result follows by selecting $m^* = m$ such that $P(A_2^c) \leq \delta$. We thus see that the probability of finding a sufficient mask occurs with probability at least $1 - \delta$. Lastly, because we have found a mask that satisfies per-parameter error $\epsilon$, and because $\epsilon \leq \varepsilon$, we have proved the lemma. $\qquad\square$

**Remark 1:** We note that, in Eq.20, $n$ will likely also depend implicitly on $\gamma$. If $\gamma$ is very small then we will need to stack many ReLUs on top of each other to attain a large enough dynamic range to approximate the desired function (see §2.1), leading to a larger number of units. Conversely, if $\gamma$ is very large we will need to sample a large number of units before we get parameters appropriately close to the desired subnetwork configuration. This suggests the existence of some sweet spot in the value $\gamma$, which we leave for future work to explore.

**Remark 2:** We observe that this bound appears to be very weak. For example, if one wished to use it to find a masked network to match an MLP with input, hidden, and output dimensions of only $1$, $3$, and $1$ respectively, with a per-parameter error of $\epsilon = 0.05$ an error probability of $\delta = 0.1$, and with $\gamma = 0.1$, this bound would suggest we need a hidden layer of $m \geq 8.34 \times 10^{12}$ neurons in the bias learning network. In light of the numerical experiments, it is clear that while the math here provides proofs of existence for bias learning it massively over-estimates the layer widths required in practice, and thus does not say anything useful about the hidden layer scaling required by bias-learning.

For the following proposition we consider the discrete time dynamical system that we wish to approximate to be as in Eq.3.

**Proposition 3.** *For finite $0 < T < \infty$, $\epsilon > 0$, and any $|\alpha| < \infty, \beta > 0$ we can find an RNN of the style of Eq. 2 of hidden width $n \in \mathbb{N}$ and a continuous mapping $r_0 : U_z \mapsto \mathbb{R}^n$ for the initial value of the RNN such that:*

$$\sup_{z_0 \in U_z, x_{0:(T-1)} \in U_{\vec{x}}} \sum_{t=1}^{T} ||\hat{y}_t(r_0(z_0), x_{0:(T-1)}) - y_t(z_0, x_{0:(T-1)})|| < \epsilon \tag{21}$$

*where $\hat{y}_t$ is the output of the RNN and $y_t$ is that of the dynamical system.*

The main portion of this result is well known, see e.g. Schäfer & Zimmermann (2006). For completeness, we provide an example proof below.

*Proof.* In what follows, W.L.O.G we will assume that the error is smaller than $c$. We want to approximate the dynamical system:

$$z_{t+1} = F(z_t, x_t), \quad y_t = Cz_t, \quad z_0 \in U_z, \tag{22}$$

defined, by assumption, on set $\tilde{U}_z = \{z_0 + c_0 : z_0 \in U_z, ||c_0|| < c\}$, where $U_z$ is an invariant set (see §2.2).

We define the set:

$$U_{zx} = \{[z + c_0 \ x] : z \in U_z, x \in U, ||c_0|| < c\}. \tag{23}$$

Importantly, this set is compact given the compactness assumptions on $U$ and $U_z$. Also note that, since $F$ is assumed continuous, it will be $K_F$-Lipschitz on this compact set for some constant $K_F$. By the compactness just discussed and the continuity of $F$, we can use the corollary to Proposition 2 to find a neural network of hidden dimension $n \in \mathbb{N}$ that approximates $F$ with a maximum-rank output matrix, $A$. We write this neural network:

$$\hat{z} = \alpha z + \beta A\phi(Wz + Bx + b) = \hat{F}(z, x), \tag{24}$$

assuming $z \in U_z$ and $x \in U$, with $A \in \mathbb{R}^{s \times n}$, $W \in \mathbb{R}^{n \times s}$, $B \in \mathbb{R}^{n \times d}$, and $b \in \mathbb{R}^n$. In particular, we can find arbitrary $\epsilon$ with $0 < \epsilon < c$ such that:

$$||\hat{F}(z, x) - F(z, x)|| < \varepsilon = \frac{\epsilon}{T \max(R_C \sum_{t=0}^{T-1} K_F^t, 1)}, \tag{25}$$

where $R_C = ||C||$. Fix $T \geq 1$. To prove that we can approximate the underlying dynamical system, we use induction starting at time $t = 1$. The base case will be

$$||\hat{z}_1 - z_1|| = ||\hat{F}(z_0, x_0) - F(z_0, x_0)|| \leq \varepsilon, \tag{26}$$

by our choice of $n$ and initial condition, and that $[z_0, x_0] \in U_{zx}$. Importantly, this implies also that $||\hat{z}_1 - z_1|| < \varepsilon$. Because $\varepsilon < c$ this means that $[\hat{z}_1 \ x_1]^\top \in U_{zx}$.

For $t = 1$, $\varepsilon = \sum_{t'=0}^{t-1} K_F^{t'} \varepsilon$. We thus make the induction hypothesis that $||\hat{z}_t - z_t|| < \sum_{t'=0}^{t-1} K_F^{t'} \varepsilon$ and that $[\hat{z}_t \ x_t]^\top \in U_{zx}$. If $T = 1$ we are finished. If $T > 1$ we assume $1 < t < T$ and use this hypothesis to prove the induction step:

$$||\hat{z}_{t+1} - z_{t+1}|| \leq ||\hat{F}(\hat{z}_t, x_t) - F(\hat{z}_t, x_t)|| + ||F(\hat{z}_t, x_t) - F(z_t, x_t)|| \tag{27}$$

$$\leq \varepsilon + K_F ||\hat{z}_t - z_t|| = \varepsilon \sum_{t'=0}^{t} K_F^{t'} < \frac{c}{T}. \tag{28}$$

Because $\frac{c}{T} \leq c$, $[\hat{z}_{t+1} \ x_{t+1}]^\top \in U_{zx}$. Then

$$\sum_{t=1}^{T} ||\hat{y}_t - y_t|| \leq R_C T \varepsilon \sum_{t=0}^{T} K_F^t \leq \epsilon. \tag{29}$$

While we have approximated the dynamical system it is not yet in the standard rate-style RNN form. However, we can obtain the rate form by changing from tracking $\hat{z}$ to a different dynamical variable, $r_t \in \mathbb{R}^n$, that satisfies $\hat{z}_t = A r_t$. We will make a brief detour to characterize this variable.

Because $A$ is full rank, $\text{col}(A) = \mathbb{R}^s$ so we can find an index $\nu \subset \{1, \ldots, n\}$ such that $\{A_{:i}\}_{i \in \nu}$ forms a basis for $\mathbb{R}^s$. If we construct a matrix $A_\nu \in \mathbb{R}^{s \times s}$ whose columns are simply the basis vectors this matrix will have an inverse $A_\nu^{-1}$. We can then define the initial condition, $r_0$, element-wise as:

$$r_{0i}(z_0) = \begin{cases} A_\nu^{-1}{}_{i:} z_0 & i \in \nu \\ 0 & i \notin \nu. \end{cases} \tag{30}$$

This function is clearly continuous and satisfies $A r_0 = \sum_{i \in \nu} A_{:i} r_{0i} = \sum_{i \in \nu} A_{:i} A_\nu^{-1}{}_{i:} z_0 = A_\nu A_\nu^{-1} z_0 = z_0$. If we then define $r_t = \alpha r_{t-1} + \phi(W A r_{t-1} + B x_{t-1} + b)$ for all $1 < t \leq T$ we see that $r_t$ is simply the state variable for an RNN of the style of Eq.2, and that it satisfies $\hat{z}_t = A r_t$ for all $0 \leq t \leq T$. It follows that $\hat{y}_t = C A r_t \ \forall t$. Thus, this RNN approximates the original partially observed dynamical system in the sense described in section §2.2.

$\square$

## C   METHODS FOR NUMERICAL RESULTS SECTION

### C.1   METHODS FOR SECTION 3.1

In this section all networks were single hidden layer FNNs. For figure 1A, networks with widths of 64, 128, 256, 512, 1024, 2048, 4096, 8192, 16384, and 32768 were trained, with the width being indicated on the $x$-axis. For figures 1B and 1C, a width of $3.2 \times 10^4$ was used.

For figure 1A, all networks were trained on FashionMNIST, with $5 \times 10^4$ training samples and $10^4$ test samples, using ADAM with a learning rate of 0.01. Training was run for 20 epochs with a batch size of 512.

Xavier uniform initialization with a gain of 1.0 was used for the fully trained networks, while the frozen weights for the bias-only networks were sampled from either a uniform distribution on $[-0.1, 0.1]$ or from a zero-mean Gaussian with standard deviation $\frac{1}{\sqrt{d}}$, where $d$ is the input dimension.

For Fig.E.3 a network with $3.2 \times 10^4$ units was trained on $2 \times 10^5$ random samples from a concatenation of the 8 MNIST style datasets used in Figure 1. Weights were initialized to Gaussians with mean 0 and SD $\frac{1}{\sqrt{d}}$, biases were initialized uniformly on $[-0.01, 0.01]$, and ADAM with a learning rate of $1 \times 10^{-5}$, and batch size 512 was used to optimized a cross entropy loss for 15 epochs. The network reached test accuracies of 0.9924 (Ethiopic) 0.8624 (Fashion), 0.8883 (Kannada), 0.0000 (KMNIST), 0.9354 (MNIST), 0.9995 (Nko), 0.9979 (Osmanya), and 0.9910 (Vai). Thus, it failed to learn KMNIST, which we hypothesize to be due to interference with the other datasets. In this figure and in Fig.1 we used the Kmeans from Scikit-Learn with 20 clusters. Before clustering/plotting we normalized the variances by dividing each 8 dimensional vector of unit variances by the maximal value plus a small constant to avoid division by zero.

## C.2 METHODS FOR SECTION 3.2

In this section all networks were single hidden layer FNNs with $10^4$ ReLU units. Weights were each initialized to a Gaussian with mean 0 and standard deviation of $1/28$–the inverse square root of the input dimension–and were left unchanged during training.

Both bias and mask networks were trained on MNIST, with $5 \times 10^4$ training samples and $10^4$ test samples, using ADAM with a learning rate of 0.01. Trained parameters were each initialized uniformly on $[-0.01, 0.01]$ (for the mask learned networks the bias vector was initialized in this way and left untrained), and training was run for 30 epochs with a batch size of 512. For bias learning, the trained parameters were the $10^4$ element bias vector, $b$; for mask learning, trained parameters were a $10^4$ element vector of gains $g$.

The output of the $i^{th}$ unit in the mask-trained network was $\text{ReLU}(W_{i:} \cdot x + b)\,\text{Sigmoid}\left(\frac{g_i}{\tau}\right)$. Here, $x$ was the input, flattened, MNIST image, $W$ the random hidden weight matrix and $\tau$ an annealing parameter. Starting from $\tau_0 = 1$, at each epoch $\tau$ was decreased so that at epoch $k$ $\tau_k = c\tau_{k-1}$, with $c$ chosen such that at the final epoch $\tau_{30} = 0.001$. This scheme was chosen so that by the end of training the sigmoid functions would be almost step functions, well-approximating the desired binary mask. We found that this worked better than immediately setting $\tau = 0.001$. For testing, units were evaluated with the Sigmoid factor binarized ($\tau = \infty$).

The null model used in Fig.2.B was calculated by drawing $10^4$ samples from two, independent, Bernoulli random variables whose probability of being ON (drawing a 1) were equal to the mean fraction of ON units in the bias and mask-trained networks respectively.

## C.3 METHODS FOR SECTION 3.3

### C.3.1 COSINE GENERATION (FIG. 3A-B)

We used $N = 200$ hidden recurrent units with ReLU activations. Biases were initialized from $\mathcal{U}(0, 1)$. The recurrent weights were sampled from $\mathcal{N}(0, N^{-1})$ and the output weights from $\mathcal{N}(0, N^{-2})$. The target period of the cosine was 25 and the total duration to generate was 125. Learning rates were 0.1 for bias learning and 0.001 for the fully-trained network; other parameters for the Adam optimizer were left at their default values in Pytorch.

### C.3.2 VAN DER POL OSCILLATOR (FIG. 3C-D)

We used 675 hidden ReLUs for the bias-learning network and 25 for the fully-trained network. The recurrent weights were sampled from $\mathcal{N}(0, gN^{-1})$, where $g$ is the "gain"; other weights were initialized as in the previous subsection. The Van der Pol oscillator obeyed $\ddot{x} = \mu(1 - x^2)\dot{x} - x$, for $\mu = 2$. Learning rates were 0.1 for bias learning and $10^{-4}$ for the fully-trained network and other optimization parameters were as above.

## C.4 Methods for Section 3.4

### C.4.1 RNN architecture and hyperparameters

The architecture is a single-layer one-dimensional input/output recurrent neural network with ReLU activations. The training optimizer was Adam with a learning rate of 0.001 and a weight decay value of 0.1.

### C.4.2 Lorenz attractor

The equations used to generate the partially-observed Lorenz attractor were the following:

$$\frac{dx}{dt} = \sigma(y - x), \quad \frac{dy}{dt} = x(\rho - z) - y, \quad \frac{dz}{dt} = xy - \beta z$$

where the initial point is $(0, 1, 0)$ and $\sigma$, $\rho$, and $\beta$ are 10, 28, and $\frac{8}{3}$, respectively. The trajectory was generated using Euler's method with a step size of 0.01.

## C.5 Methods for Section 3.5

We used $N = 1{,}024$ hidden ReLU units in a vanilla RNN (Eq. 2 with $\alpha = 0$ and $\beta = 1$). The input to the network was the 2D Cartesian position of the targets, which were 0.07 m away from the center of the workspace. The point-mass arm obeyed a discrete-time version of the following dynamics

$$\frac{dx}{dt} = \dot{x}(t), \quad \frac{d\dot{x}}{dt} = \frac{f(t)}{m}, \quad \frac{df}{dt} = \frac{u(t) - f(t)}{\tau_f}$$

where $x$ is the 2D arm position, $\dot{x}$ its velocity and $f$ the applied force. The initial conditions were $x(0) = \dot{x}(0) = f(0) = 0$, i.e., the arm was initially at rest. The force was obtained from an exponential filtering of the controls $u(t)$ with time constant $\tau_f = 0.04$ s. The network generated the controls $u(t)$ via a linear readout of its activity: $u = Cr$. The input ($B$), recurrent ($W$) and output ($C$) matrices were initialized as: $B_{ij} \sim \mathcal{U}(-1/\sqrt{2}, 1/\sqrt{2})$, $W_{ij} \sim \mathcal{N}(0, 1.5625/N)$, and $C_{ij} \sim \mathcal{N}(0, 0.5/N)$. The loss function was

$$L = \sum_{k=0}^{K-1} \frac{\|x_T^{(k)} - d_k\|^2}{\delta_p^2} + \gamma_v \frac{\|\dot{x}_T^{(k)}\|^2}{\delta_v^2} + \gamma_f \frac{\|f_T^{(k)}\|^2}{\delta_f^2}, \tag{31}$$

where $d_k = D[\cos(2\pi k/K), \sin(2\pi k/K)]^\top$, $k = 0, \ldots, K-1$, represent the position of the $K = 6$ peripheral targets at a distance $D = 0.07$ m from the center target. Here, $x_T^{(k)}$, $\dot{x}_T^{(k)}$ and $f_T^{(k)}$ are the final position, velocity and force for the $k$th target ($T = 1\text{s}/0.01\text{s} = 100$, where 0.01 s was our integration time step). Therefore, the objective was to reach the target at the end of the trial (first term) with near-zero velocity (second term) and force (third term). Parameters $\delta_p = 0.01$, $\delta_v = 0.02$ and $\delta_f = 0.08$ were used to rescale the position, velocity and acceleration terms. Hyperparameters $\gamma_v = 0.2$, and $\gamma_f = 0.04$ controlled the relative weight of the velocity and force costs with respect to the position loss. The learning rate of the standard Adam optimizer was set to $3 \times 10^{-3}$ and training was stopped when a loss of $5 \times 10^{-3}$ was reached.

## D Training Time

We ran a brief experiment on training time, which we report as $mean \pm 1SD$ over 5 seeds. Training a fully-trained single-layer MLP with $10^4$ hidden units on MNIST for 5 epochs takes $69.17 \pm 0.22$ seconds. Training only biases for the same task and architecture takes $61.51 \pm 0.92$ seconds. Thus, for matched layer widths training biases with standard Pytorch implementations provides a slight advantage. Note that one can maintain performance and get faster training by simply reducing the width of the fully-trained network: a 100 hidden unit fully trained net takes $37.77 \pm 0.04$ on the same test.

## E Supplementary Figures

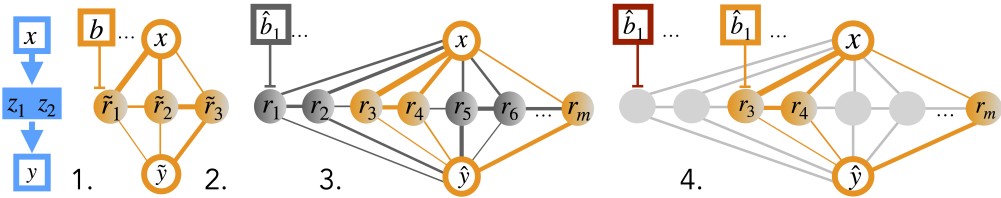

Figure E.1: **Proof Intuition. Step 1.** Consider a smooth, partially observed Dynamical System (DS) on an invariant set, $\Omega$ (or a function $y(x)$ on a compact set $\Omega$). **Step 2.** Let $0 < t \leq T$. Approximate DS on $\Omega$ using RNN $\mathcal{R}_1$ (or approximate $y$ on $\Omega$ using FNN $\mathcal{N}_1$). **Step 3.** Randomly initialize a larger RNN, $\mathcal{R}_2$ (or FNN $\mathcal{N}_2$). Check for a sub-network of units with parameters matching $\mathcal{R}_1$ ($\mathcal{N}_1$). **Step 4.** Adjust biases outside the sub-network to be very negative–thus shutting off the units outside the sub-network–and those inside the sub-network to match those of $\mathcal{R}_1$ ($\mathcal{N}_1$).

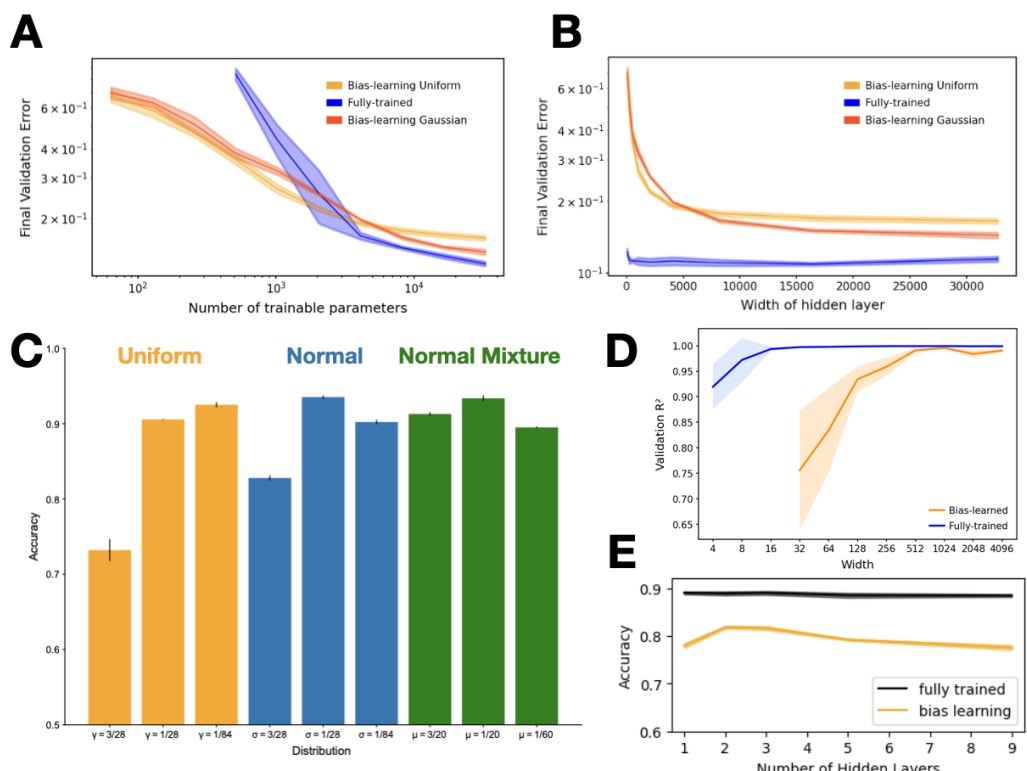

Figure E.2: **Supplementary Experiments. A.** As in Fig.1.A except with $x$ and $y$-axes log-scaled. **B.** As in Fig.1.A except $x$-axis is layer width. **C.** Performance on MNIST (y-axis) of bias learned MLPs as function of distribution from which each individual weight is initialized (x-axis). Orange: uniform on $[-\gamma, \gamma]$; Blue: 0 mean Gaussian with standard deviation $\sigma$; Green: mixture of Gaussians centred at $\pm\mu$ each with standard deviation 0.015. Layer width is $10,000$ units. **D.** As in Fig.4.A except $x$-axis is layer width. **E.** Performance ($y$-axis) of fully trained versus bias learned (uniform initialized weights) MLPs on fashion MNIST as a function of depth ($x$-axis). Layer width is 2048 units for both learning types. All plots: plotted are means and $\pm$SD over 5 random seeds.

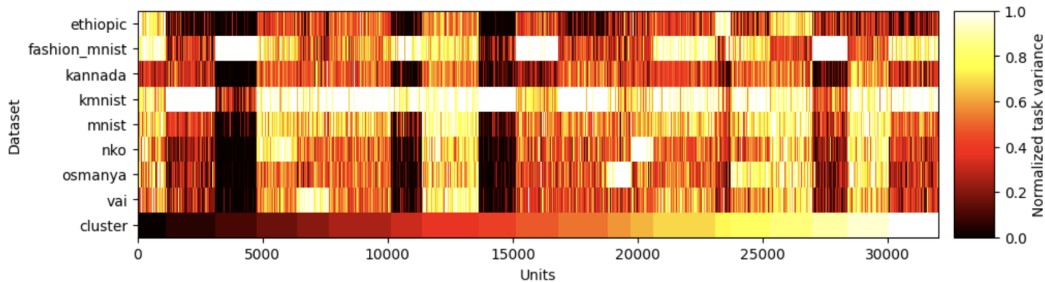

Figure E.3: **Supplementary for Figure 1.** Figure 1.C but for a single fully-trained network fit to a meta-dataset containing the 8 MNIST style datasets used in Fig.1. The bottom row is cluster index. We observe task selectivity that is qualitatively similar to bias-learning. See §C.1 for methods.

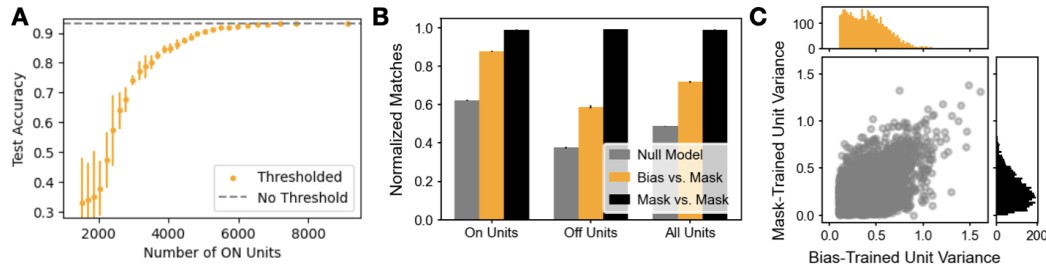

Figure E.4: **Supplementary for Figure 2. A.** Decrease in bias learning test accuracy as the task variance threshold, above which units are ON, is increased. $x$-axis is number of ON units; $y$-axis is test accuracy on MNIST. Threshold values are those used in Fig.2.B. Plotted is mean and SD over 3 training runs. **B.** Probability that a given unit is ON/OFF for versions of a network with the same weights trained by bias versus mask learning. Left group of bars is probability of the unit being ON for bias learning given that it is ON for mask learning, middle is the same but for OFF instead of ON, right is the probability of either match occurring (OFF or ON in both training paradigms). Grey is a null model (see section C.2 for details); orange is the bias-mask comparison; black is the same comparison but between two mask-learning training runs. **C.** Scatter plot of hidden unit variances taking only units that are ON for both mask and bias learning; bias-trained on x-axis and mask-trained on y-axis. Error bars are over 5 training runs. We observe a mean correlation coefficient of $0.5030 \pm 0.0087$. We note that the correlation between two mask-learning runs on the same random weights is $0.9999 \pm 0.0000$. Correlation coefficients are over 5 samples; plot data is from one, representative, pair of bias/mask networks.

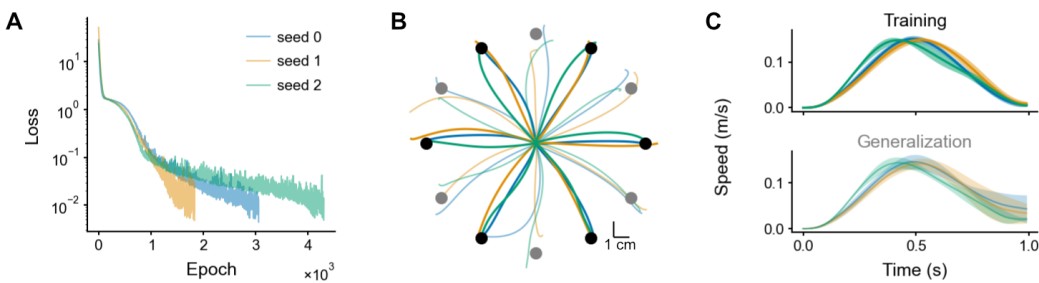

Figure E.5: **Supplementary for Figure 5.** Same figure as Fig. 5 in the main text, but for fully-trained networks. The architecture and initial parameters (seeds) were the same as for the bias-learning networks (see Methods). The only change was that the learning rate was $10^3$ times smaller than that for bias learning. For these matched parameter initializations (in particular, given the high variance for the recurrent weights), the solutions found by the fully-trained network had more temporal variations compared to bias learning.

