# OpenReview forum: "Expressivity of Neural Networks with Random Weights and Learned Biases"
_ICLR.cc/2025/Conference — ICLR 2025 Poster_

### Official Review · Reviewer_WzyV · 2024-11-02

**Soundness:** 3
**Presentation:** 4
**Contribution:** 3
**Rating:** 6
**Confidence:** 4

**Summary:**

Note: I’m not an expert in the field of pruning and universality.
The authors show that random neural networks with trained biases are universal approximators. This is shown both for feedforward and recurrent networks. The authors use an argument that is similar to masking, and use negative biases to implement the masking. Numerical simulations show that several benchmarks reach similar performance when training biases or training weights as well.

**Strengths:**

The questions of pruning, masking and universality are all important questions for neuroscience and machine learning. In neuroscience, it is known that many cell-autonomous adaptation mechanisms exist (e.g., spike threshold adaptation), and the wide distributions of firing rates hint that these properties could be a form of long-term plasticity as well.
Proving universality results has opened the door for more application-oriented research in the past. The combination of mathematical proofs with systematic simulations and a wide literature review is a strength.

**Weaknesses:**

My main concern is the scaling of network size, which seems exponential and is also missing from the main text. The results of Malach et al 2020 suggest that masking is weaker than weight-pruning, unless the size of the network is exponential.
If I understand correctly, I expect a scaling of (R/eps)^n. Because there are n independent event with probability (R/eps). Line 944 (appendix) states a scaling, which by approximating log(1+z)=z is indeed (1/eps)^(n^2).
Given this large scaling, and the results of Malach et al that with polynomial scaling neuron-pruning is weak, it seems strange that bias learning is as strong as weight learning. Indeed – the actual numerical results do not show comparable performance. As the tasks become harder, the gap widens. Also when controlling for the number of learned parameters, bias learning is still weaker.

**Questions:**

1.	Definition 2: bounded by gamma?
2.	Line 186 large hidden layer width. Can you provide a rough estimate? What is the scaling? I assume it is roughly (R/eps)^n. In Malach et al, the size was polynomial. If I understand correctly, line 944 gives such scaling, and by approximating log(1+z)=z it is indeed (1/eps)^(n^2).
3.	Fig E2. It is hard to compare what happens from 5000 parameters or so. Perhaps a logarithmic or ratio plot would help.
4.	Fig E2 – fully trained was still better than bias-only, even when controlling for parameter number. Do you know why this is? Is this related to the main concern raised above?
5.	Line 304. Correlation between TV and bias. Was this computed for every unit, and then averaged within each cluster?
6.	Line 304 If the theory is aligned with training, then the number of units should be much higher than simply the square of fully trained network. If Figure E2 suggests otherwise, then why expect the mechanism of the proof to hold? Further – what are the values of biases? Are some of them extremely low – effectively shutting down neurons?
7.	Correlation values between mask and bias – what is the correlation between different realizations of the training process?
8.	Line 461 – similar scaling in RNN and FNN. Figure E2C shows that the fully trained saturates at 64^2 parameters. The bias trained network is shown up to 64^2 parameters, so we can’t see whether fully-trained RNNs with less than 64^2 parameters behaves similarly to FNN.
9.	Line 462 stability in larger windows. Perhaps I didn’t understand this, but is this really stability or simply more test sets? Because the network is fed the true dynamics, is there a meaning to larger windows?
10.	Figure 4C – How does the fully trained network generalize in this scenario?
11.	Line 481 – I think some discussion of scaling should go into the main text, even if the proof is in the appendix.
12.	Line 483 – task-selective clusters similar to fully trained. This was not shown. Specifically, quantification of how task-selective are bias-trained vs. fully-trained networks.

---

> ### Author Response · Authors · 2024-11-22
> **Author Comment Regarding Weaknesses**
>
> We respond point-wise to questions and comments, below.
>
> # Weaknesses
> We split this response into responses to the two main sub-comments in the weakness section:
> - We are grateful for the precise feedback. You are correct that the proof uses an approach that leads to a scaling that is exponential in $n$, given the iid sampling of units. However, as we mention below (see answer to question 2), this scaling result is a very very loose estimate of the worst-case number of units needed to solve a task (as elaborated on in Remark on lines 1080-1086), and therefore not informative.
> - Regarding the Malach et al. paper: it provides a comparison between random feature networks and mask learning. Given that, for appropriate activation functions, mask learning is simply a particular parameterization of bias learning, the scaling results in Malach et al. could be interpreted as a worst case scaling for bias learning. We have added mention of this now in the manuscript (lines 515-517). A deeper quantification of the scaling properties of bias learning–whether the added flexibility of bias learning leads to better scaling than mask learning–is beyond the scope of this paper as it would require a broad array of numerical experiments or some novel theory work to properly test. However, we believe the results in our paper are relevant regardless of whether or not bias learning is ultimately found to scale better than mask learning. The reasoning is twofold: first, our empirical results show that bias learning can solve neuroscience-relevant problems with a reasonable hidden layer size, thus bias learning should not immediately be discounted as a potential learning strategy employed by the brain, or in hardware constrained settings where one may want to use a single set of weights for multiple different tasks; second, we also believe that this work provides a useful starting point, and theoretical grounding, for old and new bias-learning approaches. For example, methods, like bias fine-tuning, that could use some amount of structure in the weights intermediate between fully random (as in our results) and fully trained. Such weights would likely improve scaling, and having the expressivity results in our paper provides a theoretical backbone for this research direction.

---

> > ### Author Response · Authors · 2024-11-22
> > **Author Comment Regarding Questions**
> >
> > # Questions
> > - Yes, thank you. This is now fixed.
> > - Thank you for the detailed question! Yes, our theory yields the scaling bound in the appendix but we know that this bound is very weak–significantly under-cut by the network sizes observed in numerical results (see Remark on lines 1080-1086)–and therefore not very informative. We thus view our theory results not as a statement about scaling but as a statement of existence: for some sufficiently large, but finite, layer width one can approximate the desired function with bias learning. We leave the interesting question of results on the scaling of bias learning for the future. We also note that Malach et al.’s results are for mask learning–which may or may not provide useful insight for bias learning since, for activations like the ReLU, bias learning is more flexible than mask learning (bias learning can mask units but also translate the activation’s zero point).
> > - We have changed our feed-forward network scaling figures to log-log plots to try to remedy this; please let us know if this helps.
> > - Indeed, bias learning does not catch up to fully trained networks for the layers tested. We believe this could be due to two things (we have included mention of these in the main body of the manuscript (lines 263-266)):
> >   - Bias learning requiring a very large layer width, as suggested
> >   - Standard hyperparameter formatting being better suited for fully trained networks than bias networks
> > - No, for each cluster the correlation was computed between:
> >   - Bias for each task/unit pair
> >   - Within task variance for each task/unit pair
> > - Thank you for the feedback! Based on the numerical experiments we believe the theory overestimates the layer widths, but that the intuition about biases shutting off units is partially correct. Regarding the second question: the magnitude of a bias is not in itself a good measure of whether a neuron is shut off, as a bias may only need to be small and negative if the input weights to the given unit are sufficiently small. Instead, we have looked at task variance as a measure of whether units are ‘off’. We observe that bias learning leads to many (roughly 3000) units having a low enough task variance to not significantly contribute to performance, demonstrating many units that are functionally shut off (Supplementary Figure D3.A). We have re-worked figure 2 to better compare these ‘functionally off’ units with those in mask learning. Please let us know if this answers your questions.
> > - Could the reviewer elaborate on what is meant here? If this helps, we have also tested the correlation between two different training runs of a mask-learned network on the same weights, and observe that it is almost precisely 1 (0.99 with an almost vanishing standard deviation over 5 samples).
> > - We have added smaller layer sizes for the fully-trained model accordingly; please see the new figure for section 3.3.2 (section 3.4 in the new manuscript).
> > - We thank the reviewer for this astute observation; indeed, we realize that this is not a meaningful test of the network and have reworked the figure accordingly.
> > - Thank you for noting the lack of a fully-trained benchmark here. We have added it to the figure and observe that the performance of the fully-trained network degrades similarly to the bias-learned network.
> > - We have expanded on the discussion of scaling in the theory (see lines 180-183) and discussion sections (see lines 511-519). Thank you for the feedback; please let us know if this addresses your concern.
> > - We thank the reviewer for the intriguing idea. To calculate task selectivity for fully trained networks one would need to train a single set of weights for all 8 tasks, so that neurons can be identified and compared across tasks. The difficulty here is that one will quickly encounter catastrophic forgetting if one trains a single set of weights. Overcoming catastrophic forgetting is an on-going research topic and, thus, we believe beyond the current scope of the paper. However, such a comparison would be a fantastic future direction.

---

> > > ### Comment · Reviewer_WzyV · 2024-11-24
> > > **Author response**
> > >
> > > * I read the rebuttal and updated PDF.
> > > * Regarding my comments:
> > > * The log scale in figD.2 is easier to read.  Thanks
> > > * Figure 2 is also easier to read. Thanks.
> > > * Correlation between training runs – this is what i meant. A baseline to know how different is different.
> > > * Task-selective clusters. What I meant is quantifying the fully trained case (as in Yang et al 2019, which is cited in that section). Training all tasks simultaneously – just to see whether the degree of task selectivity is similar or not.
> > > * Regarding the new reaching task – do you have any baseline of weight training? Also - if I understood correctly, the output of the network after reaching the target is zero. Therefore, it is not clear whether there are distinct fixed points of the dynamics.

---

> > > > ### Author Response · Authors · 2024-11-25
> > > >
> > > > Thank you, we are very grateful for your attentive reviewing of our manuscript!
> > > > To address your bottom three bullet points:
> > > > - On correlation: perfect, please let us know if there is anything else we can clarify with regards to these statistics.
> > > > - On task-selectivity: thank you very much for the clarification. Our objective with task selectivity was to provide a characterization of this phenomenon for bias learning rather than a comparison across different learning paradigms. While this is an intriguing direction, we think a detailed comparison of neural representations for fully-trained vs bias-trained networks is a little outside the scope of this work. Regardless, we are setting up a task-selectivity experiment for a fully-trained MLP and will post it here if it is ready before the end of the rebuttal period.
> > > > - On the reaching task: to address this comment we ran the same experiments for a fully-trained RNN. We observed qualitatively similar results and have included them in the appendix (see Fig.D4); please let us know if you have any clarifying questions. Regarding the fixed point question, yes, you are absolutely right–the dynamics observed could be due to something else, for example dynamics that move into the nullspace of the output matrix. For this reason we didn’t reference fixed-points specifically in the updated manuscript. We did mention “fixed point-like dynamics” in another reviewer response, which we have subsequently deleted given the ambiguity that you bring up.
> > > >
> > > > Thank you again for the detailed and valuable feedback which, we believe, has improved the manuscript. Please let us know if there is any other information we can provide that might help you in judging potential changes in your rating of our paper.

---

> > > > > ### Author Response · Authors · 2024-11-28
> > > > >
> > > > > To follow-up on the last message, we have added a new figure to the appendix of the updated manuscript showing the task selectivity plot for fully-trained networks, as requested (see Figure D.3 and associated methods). Please let us know if there is any extra information we can provide to aid you in your evaluation of our paper. In the meantime, thank you again for all your feedback. We believe this review period has meaningfully strengthened our manuscript, and your insights have played a crucial role in the process.

---

> > > > > > ### Comment · Reviewer_WzyV · 2024-11-29
> > > > > >
> > > > > > I thank the authors for the modifications and responses. Regarding Figure D3, it seems that clustering is stronger in the weight-trained network, but quantifying this would require much more experiments.
> > > > > > The claims made in the rebuttal regarding a mix of synaptic and bias learning are very speculative.
> > > > > > After carefully reading all the discussion, I am raising my score to 6.

---

> > > > > > > ### Author Response · Authors · 2024-11-29
> > > > > > >
> > > > > > > Thank you very much for your support of our paper, and for your careful and detailed reviewing! We agree that it appears, qualitatively, that there is greater selectivity in the bias-learned network, and that this, while outside the scope of the current work, would represent an intriguing direction for future study. We will attempt to fit mention of this into the discussion section. We also confirm that the idea of combining simple synaptic structuring with bias learning is indeed speculative (we are actually beginning to investigate this currently!), which is why we have limited this speculation to the discussion portion of the manuscript. We believe that it is worth discussing because of the growing interest in combinations of synaptic and input-driven learning (see e.g. citations in second paragraph of revised manuscript). Regarding your support of our paper, we see that the score was not yet updated in your main review and just wanted to add a reminder about this, in case this step was forgotten, for the sake of the openreview stats. We are also happy to address any other questions or clarifications you might have that could lead you to further strengthen your support. Thank you again so much for all your help in the reworking of our manuscript!

---

> ### Author Response · Authors · 2024-12-02
> **[Addressed] IMPORTANT: adressing score mismatch between comment and system**
>
> UPDATE: Thank you for rectifying the score entered in the system. The score now reflects the intent communicated in the discussion.
>
> -----
> Hello,
>
> We noticed that the reviewer has indicated that they would raise their score (see last reviewer comment from Nov 29) but the system is currently not reflecting this change. Could it be that the score change did not get saved? As the discussion period is drawing to an end, we kindly point out this discrepency and once again thank the reveiwer for the fruitful exchange.  We remain available to adress any additional points.

---

### Official Review · Reviewer_kfuh · 2024-11-03

**Soundness:** 3
**Presentation:** 3
**Contribution:** 2
**Rating:** 6
**Confidence:** 4

**Summary:**

In this work, the authors prove a universal approximation result for the expressivity of neural networks with frozen weights but trainable biases. In particular, they build upon well-known universal approximation results of feedforward and recurrent architectures, and show via a simple mask learning-like argument that sufficiently large networks with randomly chosen weights can be constructed to approximate any function (feedforward) or finite-time trajectory (recurrent). They conduct experiments comparing fully trainable architectures to bias-learning variants, demonstrating that bias-only learning can achieve reasonable performance on some simple tasks.

**Strengths:**

The main expressivity results shown are well-explained and seem mathematically tight. Considering that these results made use of a reduction to mask learning problems, the authors also do a good job discussing the relationship between their findings and those of the mask learning literature.

**Weaknesses:**

A crucial aspect of this work with regards to its practical relevance is how large a bias-trained network needs to be to achieve similar performance to a fully trained network. Surely the scaling is better than the extreme network expansions constructed for the existence proofs, but how much better? The authors allude to performance as a function of trainable parameter count scaling similarly to fully trained networks, and thus only needing quadratic scaling in layer width, but they only evidence this explicitly with comparisons to mask learning networks, which in my view are also less expressive than standard networks (for fixed number of parameters). I would like to see a more detailed investigation of this question. For example, could the authors extrapolate from the MNIST experiments (Fig. 1a) whether the required scaling is indeed quadratic? I imagine this scaling would also depend significantly on the task difficulty and the frozen weight initialization.

Overall, the tasks the authors used to demonstrate efficacy of bias-only learning seemed restrictively simple, by the standards of both the machine learning and computational neuroscience literature. In particular, for the RNN experiments, only simple 1D pattern generation tasks were considered. I would be interested in seeing how biased-trained RNNs perform on simple "cognitive" tasks often used to assess task-trained RNNs in the computational neuroscience literature (e.g., interval timing, delayed match-to-sample). For example, I imagine that any task that requires the construction of many stable fixed points could be quite difficult for bias-learning RNNs, and might require prohibitive scaling of network size compared to fully-trained counterparts (e.g., N-bit flip flop).

**Questions:**

See above.

---

> ### Author Response · Authors · 2024-11-22
> **Author Comment**
>
> We respond point-wise to the reviewer's questions below.
>
> # Weaknesses
> - Thank you to the reviewer for the astute question. We know for certain that bias learning scales better than the extreme scaling suggested by the theory, as we can calculate the network sizes suggested by the theory and find that, numerically, bias learning scales a lot more reasonably (see remark 2 on lines 1080-1086 of the appendix for details). As such, we view the theorems as a statement that bias learning will work for some sufficiently wide layer, rather than a characterization of the precise scaling of bias learning. What then, of the key question of scaling itself? Because bias learning, with the activation functions tested, encompasses mask learning (since bias learning can shut off units), we know that the scaling will perform at least as well as masking units. Past work has related the scaling of unit masking to the scaling of random feature networks, and we have added mention of this in the new manuscript accordingly (see lines 513-517). We appreciate the idea of calculating scaling from the numerical work performed; to this end we reworked the scaling sub-figures, Fig.1A and Fig4.A, so that they directly compare performance as a function of trainable parameter count with a log-axis to better visualize the different curves. We have also plotted a log-log axis (Fig.D.A) to attempt to extrapolate the scaling, as you suggested. For power law (polynomial) scaling we would expect a linear relationship on the log-log plot, which is what we, approximately, observe. However, we hesitate to draw general conclusions from this one experiment because, as you mention, we would expect quantities like task difficulty and hyper-parameter choice to impact things. To address scaling numerically we would ideally run an array of experiments over a wider range of layer widths, a task that we believe is beyond the scope of the current work (we mention this as a future direction in the discussion–see lines 517-519). In the meantime, we believe that the reasonable performance of bias learning on the benchmarks tested render it a potentially useful neuroscientific model for rapid adaptation of network dynamics, and a step towards two lines of AI application: models that initialize weights from more structured distributions and hardware constrained settings, where one may want to use the same weights for multiple tasks, e.g. in small devices where weight change is prohibitive.
> - We thank the reviewer for the great suggestion. We found that RNNs can indeed be trained to perform neuroscience-like tasks using bias learning. We have added a classic neuroscience task: the center-out reaching task (see new section “Motor control”, starting on line 455) . This task requires the RNN to move a cursor from the center of a circle to various points on the radius (see methods section C.5 for details). We believe the reviewer’s insight here has significantly improved the manuscript, and that the new results provide deeper evidence for the relevance of this work for neuroscientific modelling.

---

> > ### Comment · Reviewer_kfuh · 2024-11-25
> >
> > I appreciate the authors' revisions, especially the inclusion of the motor control task. I still feel that a more complete investigation of required scaling in bias-only networks is called for, but the authors do a reasonable first-pass characterization here. I thus have raised my score to a 6.

---

> > > ### Author Response · Authors · 2024-11-25
> > >
> > > Thank you so much for the sound feedback you have given us; your insights have shaped our manuscript into a more worthy contribution. We are grateful that you see the value in our work as a first-pass characterization of bias learning. Please don't hesitate to inquire after any other information that would inspire you to further increase your score, and the chance of having our results on bias learning made available to the ICLR community. Thank you again!

---

### Official Review · Reviewer_jeUd · 2024-11-07

**Soundness:** 4
**Presentation:** 4
**Contribution:** 3
**Rating:** 8
**Confidence:** 3

**Summary:**

The authors show that both feedforward and recurrent neural networks can act as universal function approximators for functions and dynamical systems respectively, even when only biases are learned. They propose an alternative proof for the theorem that masking is all you need (Strong Lottery Ticket Hypothesis), and extend that and the bias result to RNNs approximating dynamical systems. They authors demonstrate their results using simple simulations, and discuss relevance to AI and neuroscience.

**Strengths:**

The authors connect well to the neuroscience and AI/ML literature and explain the proofs in an intuitive manner.
The extension to RNNs and dynamical systems is also commendable as these often receive reduced attention in the ML community.
The issue with the "gain" g in the weight distribution is well brought out.

**Weaknesses:**

The section 3.3.2 on the Lorenz system is not clearly written and the architecture and external input to the network are not clear.

At first glance, the result seems to be a simple extension of the masking theorem of Malach et al 2020.  The difference with that proof should be made clear.

**Questions:**

In Section 3.3.2 - The authors write RNN, but then say that the recurrent state is given? Also what is provided as an external input? Is it the recurrent state? The difference between the 'standard' and the 'self-sustained' networks is not clear. To me the self-sustained way is the standard, and if somehow the recurrent state is provided (at each time step?), then the network is just acting as a feedforward network. Then in this case, I suspect that to actually use the RNN (usual self-sustained way) to learn the dynamics, the authors would need a lot more units.

The authors have not explained how (positive & negative) biases may arise in neuroscience if not by synaptic weights. As they mention threhsold changes etc. change the neural gain (and possible have a strongly non-linear effect). What about the role of inhibition and other brain areas switching parts of the network on and off?

The authors should bring out the differences between their proof and the Malach et al proof.

---

> ### Author Response · Authors · 2024-11-22
> **Author Comment**
>
> We respond point-wise, to the reviewer's questions, below.
> # Questions
> - We thank the reviewer for the feedback on section 3.3.2–now section 3.4 in the new manuscript. To address these clarity issues we have written a methods section, in the manuscript appendix, for section 3.4 along with all the other numerical work. We have also adjusted the main text for clarity. In particular, regarding your questions here, we note:
>   - The recurrent state is not given in our experiments–only an input, in the non-autonomous case (no inputs are given in the autonomous case). We have tried to make this more clear in our current edits, with changes in the first paragraph of section 3.4.
>   - We apologize for the lack of clarity here. In the standard case the RNN predicts future steps of the dynamical system given past steps of a partially observed history as an input (this is the classic time series forecasting problem). In the self-sustained case the RNN receives its very own prediction of future states as the input from which it is predicting the future (here the RNN is now generating samples from the time series when it was only trained for forecasting). We have updated the text in the second paragraph of section 3.4 and in the Fig.4 caption to better communicate this.
> - We thank the reviewer for this comment which led us to expand the discussion of key neuroscience mechanisms which are typically modeled as bias modulations or bias learning (listed in a new table in supplemental section A). In particular, if we model a brain area as a local neuronal circuit, then the effects of tonic inputs from other areas onto the local circuit can be modelled by setting specific values for the biases. These biases can switch certain parts of a network on/off as in models of motor behavior. In the first model, based on a coupled thalamocortical-basal ganglia neural network, the basal ganglia projection to thalamic neurons are modeled as inhibitory biases, which turn off thalamic populations to induce specific motor patterns (Logiaco et al. Cell Reports (2021)). In an extension of this model (Recanatesi et al. Neuron (2022); Mazzucato Elife (2022)), the secondary motor cortex generates tonic inputs to the primary motor cortex, modeled as biases in the motor cortex network: each set of biases represent initial conditions for a particular action and are active for the whole duration of the action. Input-driven learning from upstream regions sending inputs to the motor cortex has also been able to explain short-timescale, trial-by-trial behavioral adaptation to predictable perturbations (Perich et al. Neuron 2018; Feulner et al. Nature Communications (2022)). In the typical neuroscience approach where biases to the local circuit represent external inputs from other brain areas, biases can be positive or negative depending on the particular neurotransmitter which mediates such long range projections (i.e., AMPA, GABA, dopamine, acetylcholine, norepinephrine, serotonin, etc). We have elaborated on this in the main text, adding reference to the new table, in lines 51-55, to address your astute feedback.
> - Thank you to the reviewer for the suggestion! We have included a more detailed description of the differences between our proof and Malach’s proof in the appendix (see lines 917-927 of the new pdf), and referenced this in the introduction (see lines 95-98). Please let us know if this is sufficient.

---

> > ### Comment · Reviewer_jeUd · 2024-11-24
> >
> > I have gone over the authors' reponse and the updated pdf. They have address my concerns satisfactorily.

---

> > > ### Author Response · Authors · 2024-11-25
> > >
> > > We are very grateful to the reviewer for the care put into reviewing our manuscript--your feedback has been invaluable in crafting a stronger paper. Please don't hesitate to let us know if there is any more information we can provide that might lead you to further increase your support of our work. Thank you so much for all the insight!

---

> > > > ### Comment · Reviewer_jeUd · 2024-11-26
> > > >
> > > > I do not think anything further in the limited amount of time will lead to increasing my score further, since I had already given an 8. Resolving the earlier unclearness(es) only confirms what I had assumed, while making the paper clearer, and so I maintain my score.

---

> > > > > ### Author Response · Authors · 2024-11-28
> > > > >
> > > > > Understood; thank you so much for the endorsement of our work, and for the time put into your detailed reviewing!

---

### Official Review · Reviewer_1PGv · 2024-11-12

**Soundness:** 3
**Presentation:** 4
**Contribution:** 2
**Rating:** 6
**Confidence:** 5

**Summary:**

Previous work has investigated the expressivity of feed-forward neural networks (FNNs) when only subsets of parameters are trained (ie. only the output layer, normalization parameters, etc…). In the same vein, the authors introduce a method of training feed-forward neural networks by randomly sampling fixed weights and subsequently learning only the biases, termed bias learning. They provide theoretical and empirical evidence that demonstrates that FNNs trained through bias learning can approximate any continuous function on compact sets - meaning that they are theoretically as expressive as fully-trained FNNs.

They start with a theoretical treatment of bias learning where they carefully define their terms and introduce their theorems. A simplified version of the rigorous proof is as follows: 1) Train a fully connected network (N1) where the weights are constrained to lie in some fixed range. 2) Create a new network (N2) by randomly sampling the hidden neuron weights from the fixed range in 1. 3) After sufficient sampling, there exists a subnetwork of neurons in N2 that is ‘identical’ to the neurons in N1. 4) By training the biases, the outputs of neurons outside of this subnetwork can be removed, leaving N1 from N2 bias training. They provide a similar proof for recurrent neural networks (RNNs).
Next, the authors provide empirical evidence supporting their theory and explore the expressivity of bias-learned networks. They do this in multiple ways, including performing multi-task learning with bias learning in 7 tasks (MNIST, KMNIST, Fashion MNIST, etc.), comparing bias learning and mask learning in FNNs, and applying bias learning on an RNN trained on both an autonomous and non-autonomous dynamical system. The main takeaways are as follows: 1) multi-task bias learning leads to emergence of task-specific functional organization revealed by clusters of activation patterns measured by task variance, 2) compared to mask learning, bias learning had less sparse solutions and higher unit variance values, 3) bias learning in RNNs can succeed in time-series forecasting of non-linear dynamical systems with high enough gains.

**Strengths:**

- Overall, there is strength in its novelty of proving that bias learning in neural networks can have high expressivity that performs almost as well as a fully-trained network. This is significant because bias learning trains fewer parameters than a full network.
- Nature of bias learning is more behaviorally relevant in the context of tonic inputs, intrinsic cell parameters, threshold adaptation, and intrinsic excitability
- The theoretical proofs are very thorough, and backed up by numerical proofs.

**Weaknesses:**

- In response to bias learning having fewer parameters to learn, no data was shown on training time
- Little background was given on mask learning (the mask learning section was also super short - felt less developed relative to other parts of the paper). This is important because two of their highlights in the results relate to mask learning.
- Makes claims (i.e. lines 253 - 259, lines 418-420) that could have been easily backed up by data, but were not.
- Figure 1 color scheme is weird
- Practically, not sure how exciting this is (i.e. other models can do what this model does - it's just that they use a different approach)
- The work seems highly related, in spirit, to neural tangent kernel approaches and other methods that consider wide NNs, but no references to that work were made.

**Questions:**

N/A

---

> ### Author Response · Authors · 2024-11-22
> **Author Comment**
>
> We respond point-wise, to the reviewer's questions and comments, below.
> # Strengths
> We thank the reviewer for the kind words about our theoretical contributions!
> # Weaknesses
> - Thank you for the observation! We do not claim that bias learning would require fewer parameters than fully trained networks–at least not with the fully random weights studied in our paper. To make this more clear we show parameter-matched performance results in the main manuscript (see Fig.1A and Fig.4A). Regarding training time, we have not focused on such efficiency comparisons because bias-learned networks with fully random weights do not train faster than fully-trained networks (a 100 unit fully-trained network will take less time to train and still perform slightly better than a 10000 unit bias-learned network) on the tasks we explored. We agree that this would be problematic if our paper’s objective was to provide out-of-the-box methods for SOTA accuracy or efficiency gains, but this is not our objective. We believe the value in our paper is in theoretical insights for models of learning in neuroscience and for future ML development; for example, as a starting point for understanding/designing bias learning algorithms that rely on weights with some amount of structure intermediate between fully random and fully trained (see discussion starting at line 528 in new manuscript).
> - We thank the reviewer for the helpful criticism. We have re-worked Figure 2 to better illustrate the solutions found by mask versus bias learning (see updated manuscript), and have edited the text accordingly (see section 3.2 of the new pdf). We have also added a methods section to the manuscript’s appendix, giving full details on the implementation of the mask learning process. Please let us know if there is anything that is still unclear with this portion of the paper.
> - We thank the reviewer for the comment. We have removed lines 418-420 because the statement, while theoretically correct, has little practical relevance for very small matrix gains. Could the reviewer elaborate on which claim in lines 253-259 could be backed up with data?
> - We have updated the color scheme of figure 1 (see new manuscript pdf).
> - We agree that taken as a novel method, this approach might seem limited. In the Common Comment above, we argue that the main contribution of this work is to establish new theoretical insights and guarantees on optimization approaches that do not target connections, but rather rely on constant inputs to units (i.e. biases). Above, we also stress the relevance of our work to computational neuroscience and in-context learning in LLMs. Overall, the fundamental expressivity of input-driven networks is not well-studied, and just like the universal approximation theorem for standard deep networks lays out idealized cases to provide foundations, our result contributes a first step in input-driven expressivity. We incorporated these points into the revised introduction.
> - Our work is fundamentally different from neural tangent kernels in that it studies wide but finite networks, rather than the infinite-width limit. However, we believe that exploring infinite-width limits of bias-learned networks could represent an exciting future direction, and we thank the reviewer for the inspiration!

---

> > ### Author Response · Authors · 2024-11-25
> >
> > We would like to follow-up to see if there is any further clarification we can provide to the reviewer that might lead them to potentially raise their score, and support the communication of our results to the ICLR community. Thank you again, very much, for the detailed review!

---

> > ### Comment · Reviewer_1PGv · 2024-11-26
> >
> > Thanks to the authors for a clear response. My most significant concerns were addressed by the authors response, and the improved manuscript (improvements mostly stemming from other reviewer's comments) has, I believe, pushed this work above the bar for acceptance. (I have changed my score from a 5 to a 6). I still believe additional work is necessary to understand how these approaches perform (e.g. training time) against fully trained networks, but I do believe introducing the approach and its impact on ideas in computational neuroscience to the ICLR community has value.

---

> > > ### Author Response · Authors · 2024-11-28
> > >
> > > Thank you very much for all the engagement, and for the kind words about our work. As a first stab at addressing your question about training time we ran some experiments, which we report as $\mathrm{MEAN}\pm 1\mathrm{SD}$ over 5 seeds. On our hardware, training a fully-trained single-layer MLP with $10^4$ hidden units on MNIST for 5 epochs takes $69.17\pm0.22$ seconds. Training only the biases for the same task and architecture takes $61.51\pm0.92$ seconds. Thus, for matched layer widths training biases with standard pytorch implementations provides a slight advantage. Note that one can maintain strong performance and get faster training by simply reducing the width of the fully-trained network (a 100 hidden unit fully trained net takes $37.77\pm0.04$ seconds on the same test). However, as mentioned (see our `common response’, or the first bullet point in our response to your detailed review), our objective with bias learning is not to present an efficient, out-of-the-box method but rather to provide theoretical insights to communities like computational neuroscience. For this reason, while these efficiency results provide extra context we don’t see them as absolutely critical given the objectives of our paper. Nonetheless, we appreciate you advocating for a test of these train-time differences, and we will add them to the supplementary section of our manuscript.
> > >
> > > If there is anything else that we can provide that might compel you to increase your support of our work, and our chances of getting to communicate it to the community, please do let us know. Thank you again for all your great feedback–your suggestions have improved the quality of our manuscript!

---

### Meta-Review · Area_Chair_y18U · 2024-12-24

**Metareview:**

This paper makes a significant theoretical contribution by demonstrating that neural networks with fixed random weights and trainable biases are universal function approximators.  The authors provide rigorous proofs complemented by empirical experiments, including tasks relevant to neuroscience and AI, such as motor control and multi-task learning benchmarks.  All reviewers voted for acceptance.

**Additional Comments On Reviewer Discussion:**

While some reviewers raised concerns about the scalability of bias-trained networks and the simplicity of the chosen tasks, the authors addressed these points through revisions that clarified theoretical bounds, expanded experimental scope, and enhanced presentation. Though the practical utility of bias learning remains speculative, its relevance to neuroscience and its potential for inspiring structured weight initialization in AI are notable strengths.

---

### Decision · Program_Chairs · 2025-01-22

Accept (Poster)